# Chromatin topology defines estradiol-primed progesterone receptor and PAX2 binding in endometrial cancer cells

Alejandro La Greca[1], Nicolás Bellora[2,3†], François Le Dily[4,5†], Rodrigo Jara[1], Ana Silvina Nacht[4,5], Javier Quilez Oliete[4], José Luis Villanueva[4], Enrique Vidal[4], Gabriela Merino[6], Cristóbal Fresno[6], Inti Tarifa Reischle[1], Griselda Vallejo[1], Guillermo Vicent[4], Elmer Fernández[3,6], Miguel Beato[4,5], Patricia Saragüeta[1,3*]

[1]Biology and Experimental Medicine Institute, IBYME-CONICET, Buenos Aires, Argentina; [2]Institute of Nuclear Technologies for Health, INTECNUS-CONICET, Bariloche, Argentina; [3]Consejo Nacional de Investigaciones Científicas y Técnicas (CONICET), Buenos Aires, Argentina; [4]Centre for Genomic Regulation (CRG), Barcelona Institute for Science and Technology, Barcelona, Spain; [5]Universitat Pompeu Fabra (UPF), Barcelona, Spain; [6]Bioscience Data Mining Group, Córdoba University, Córdoba, Argentina

**\*For correspondence:**
patriciasaragueta2@gmail.com

[†]These authors contributed equally to this work

**Competing interest:** The authors declare that no competing interests exist.

**Abstract** Estrogen (E2) and Progesterone (Pg), via their specific receptors (ERalpha and PR), are major determinants in the development and progression of endometrial carcinomas, However, their precise mechanism of action and the role of other transcription factors involved are not entirely clear. Using Ishikawa endometrial cancer cells, we report that E2 treatment exposes a set of progestin-dependent PR binding sites which include both E2 and progestin target genes. ChIP-seq results from hormone-treated cells revealed a non-random distribution of PAX2 binding in the vicinity of these estrogen-promoted PR sites. Altered expression of hormone regulated genes in PAX2 knockdown cells suggests a role for PAX2 in fine-tuning ERalpha and PR interplay in transcriptional regulation. Analysis of long-range interactions by Hi-C coupled with ATAC-seq data showed that these regions, that we call 'progestin control regions' (PgCRs), exhibited an open chromatin state even before hormone exposure and were non-randomly associated with regulated genes. Nearly 20% of genes potentially influenced by PgCRs were found to be altered during progression of endometrial cancer. Our findings suggest that endometrial response to progestins in differentiated endometrial tumor cells results in part from binding of PR together with PAX2 to accessible chromatin regions. What maintains these regions open remains to be studied.

## Editor's evaluation

This study modeling the actions of estrogen and progesterone receptors (ER and PR) in endometrial cancer cells through a panel of genomic approaches reveals a potentially interesting collaboration between the two, further facilitated by the non-receptor transcription factor PAX2. The identification of so-called chromatin 'progestin control regions' inside TADs, where the three factors cooperate and which appear to be the feature setting endometrial cancer cells apart from breast cancer cells, is of potential interest for future investigation.

## Introduction

Progesterone (Pg) is a key regulator in the female reproductive tract, including uterine and mammary gland development (**Lydon et al., 1995**). Endometrial and breast tissues exhibit significantly different responses to hormones, resulting in very distinctive morphologies and functions. During pregnancy, Pg prepares the uterine epithelium to receive the embryo and initiates the process of differentiation of stromal cells toward their decidual phenotype. In the mammary gland and in coordination with prolactin, Pg stimulates epithelial proliferation and differentiation of alveolar lobes in the mammary gland (**Mulac-Jericevic and Conneely, 2004**). Unlike Pg, estradiol (E2) is the main proliferative signal in the uterine epithelium and exerts its function through activating estrogen receptor (ER) alpha and beta (ERalpha and beta, respectively) (**Ishiwata et al., 1997**; **Kayisli et al., 2004**).

The physiological role of Pg is mediated by the interaction and consequent activation of isoforms A (PRA) and B (PRB) of the progesterone receptor (PR), which are transcribed from alternate promoters of the gene (**Hovland et al., 1998**). While PRA is more abundant in stromal endometrial cells, PRB is the most representative isoform in ephitelial cells of endometrium. Steroid hormones exert their transcriptional effects through binding of the steroid receptors (SR) to specific DNA sequences in the promoters or enhancers of target genes known as 'hormone response elements' (HRE). Estradiol exposure triggers ER binding to estrogen response elements (ERE) regulating target genes such as *PGR*. Previous work showed E2-dependent upregulation of PR in many different target cells, species and pathological conditions (**Graham et al., 1995**; **Kraus and Katzenellenbogen, 1993**). Exposure to progestins triggers binding of PR to PRE. Once bound to their HREs the hormone receptors interact with other transcription factors, co-regulators (**Beato et al., 1995**), such as the p160 family of co-acti-vators of steroid receptors SRC-1–3, and chromatin remodeling enzymes. This evidence favors tissue specific roles of PR isoforms and their co-regulators orientated towards differential transactivation of target genes.

High levels of PRA and PRB have been described in endometrial hyperplasia (**Miyamoto et al., 2004**) while low- and high-grade endometrial cancers reveal reduced or absent expression of one or both isoforms in epithelia or stroma (**Shao, 2013**). This PR decrease is often associated with shorter progression-free survival and overal survival rates (**Leslie et al., 1997**; **Miyamoto et al., 2004**; **Saka-guchi et al., 2004**; **Jongen et al., 2009**; **Kreizman-Shefer et al., 2014**). The absence of PR gene expression may be attributed to hypermethylation of CpG islands within the promoter or first exon regions of the PR gene or to the presence of associated deacetylated histones. These modifications were reported for endometrial cancer cell lines as well as tumor samples and may be exclusive to PRB (**Sasaki et al., 2001**; **Xiong et al., 2005**; **Ren et al., 2007**). Treatment of such cells with DNA meth-yltranferase or histone deacetylase inhibitors can restore both PRB expression and its regulation of target genes such as *FOXO1*, p21 (*CDKN1A*), p27 (*CDKN1B*), and cyclin D1 (*CCND1*) (**Xiong et al., 2005**; **Yang et al., 2014**). Down-regulation of PR by post-transcriptional mechanisms and through post-translational modifications of PR may contribute to progesterone resistance in endometrial cancer but have not been extensively explored in the context of endometrial cancer. It is known that oncogenic activation of KRAS, PI3K, or AKT and/or loss of functional tumor suppressors such as *PTEN* are common genetic alterations (**Hecht and Mutter, 2006**), toghether with *ARID1A* (**Liang et al., 2012**), all of them observed in endometrial cancer. Although there are numerous reports of hormon-ally regulated enhancers and super-enhancers in mammary cancer cells (see in dbsuperenahncer, http://bioinfo.au.tsinghua.edu.cn/dbsuper/) (**Khan and Zhang, 2016**; **Hnisz et al., 2015**), there is a void of information about their presence in endometrial cells.

To better understand the response to progestin in endometrial cancer cells, we have studied the genomic binding of ER and PR, the global gene expression changes and the state of chromatin by ATACseq as well as the genomic interactions by Hi-C in Ishikawa cells exposed to progestin or estrogen, and also in cells exposed to progestin after a period of estradiol pretreatment. Inside TADs with progestin regulated genes, we identified regions that we named 'progestin control regions' (PgCRs) that correlate with the open chromatin compartment independently of hormonal stimuli and include binding sites for the partner transcription factor PAX2.

# Results

## Ishikawa endometrial epithelial cells respond to R5020 through activation of PR, whose levels increase upon exposure to E2

Endometrial epithelial cells respond to ovarian steroid hormones -progesterone (Pg) and estradiol (E2)-, E2 being the main proliferative stimulus and Pg its antagonist. After treating Ishikawa cells with E2 10 nM for 48 hr we observed an increment in number of cells compared to vehicle (OH) (FC 1.78±0.08 v. OH) that was suppressed by addition of R5020 10 nM (FC 1.15±0.08 v. OH) (*Figure 1A*). Treatment with R5020 10 nM alone did not induce proliferation on Ishikawa cells (FC 0.77±0.08 v. OH) (*Figure 1A*; *Figure 1—source data 1*). E2-induced cell proliferation was also abrogated by pre-incubation with estrogen receptor (ER) antagonist ICI182780 1 μM (ICI $10 \times 10^{-6}$ M) (FC 1.05±0.05 v. OH) (*Figure 1—figure supplement 1A*), but not by pre-incubation with PR antagonist RU486 1 μM (RU486 $10 \times 10^{-6}$ M) (FC 1.42±0.07 v. OH) (*Figure 1—figure supplement 1B*), proving that ER but not PR was directly involved in the proliferative response to E2. Suppression of E2-induced cell proliferation by R5020 was inhibited by pre-incubation with RU486 (FC 1.50±0.06 v. OH), indicating that R5020 effect was mediated by PR in Ishikawa cells (*Figure 1—figure supplement 1B*). The effects of E2 and R5020 on proliferation were corroborated by BrdU incorporation and cell cycle phase analysis 18 hr after hormone exposure (*Figure 1—figure supplement 1C and D*). E2 increased the number of BrdU positive cells and percentage of cells in S phase compared to untreated control cells and to cell exposed to the vehicle (OH), and these increments were inhibited by R5020. Treatment with the histone deacetylase inhibitor Trichostatin A 250 nM (TSA 250 nM) was used as negative control for BrdU incorporation and cell cycle progression (*Figure 1—figure supplement 1C*).

Ishikawa cells contain isoforms A and B of PR (PRA and PRB), both of which increased their steady state levels by treating cells with E2 10 nM for 12 h (*Figure 1—figure supplement 1E and F*). Pretreating cells with E2 for 12 h (preE2) followed by R5020 had little effect on the proliferative response (*Figure 1A*), while E2 pre-treatment for 48 h significantly increased the proliferative effect of E2 exposure compared to non-pretreated cells (FC 1.47±0.08 v. no preE2). The percentage of cells exhibiting nuclear localization of PR increased upon E2 pretreatment prior to R5020 exposure (T0). Upon exposure to R5020 for 60 min the percentage of cells exhibiting nuclear PR was not affected by E2 pretreatment, though the intensity of the fluorescence signal increased in E2-pretreated cells (*Figure 1B*; *Figure 1—source data 2*). Ishikawa cells express considerably higher levels of ERalpha than of ERbeta (*Figure 1—figure supplement 1G*), suggesting that the proliferative effect of E2 was mediated by ERalpha. R5020 increased nuclear ERalpha, suggesting a functional PR-ER crosstalk in response to hormonal stimuli (*Figure 1—figure supplement 1H*). Such interactions have already been proven in breast cancer T47D cells (*Ballaré et al., 2003*) and in UIII rat endometrial stromal cells (*Vallejo et al., 2005*), though in the latter PR remains strictly cytoplasmic.

Treatment with hormones during 12 h produced transcriptomic changes consistent with the physiological stages of normal cycling endometrial tissue (*Chi et al., 2020*). RNAseq results from Ishikawa cells (*Figure 1—source data 3*) exposed to E2 10 nM for 12 h showed a significant resemblance to proliferative endometrium (*Figure 1C*), while 12 h treatments with R5020 10 nM regulated a gene expression profile similar to a mid-secretory phase (*Figure 1D*). In line with these findings, among the top overrepresented biological processes for E2-treated Ishikawa cells showed angiogenesis and positive regulation of smooth muscle cell proliferation and for R5020-treated cells processes like protein targeting to Golgi and SRP-dependent cotranslational protein targeting to membrane were found (*Figure 1—figure supplement 2A*). In addition, the majority of regulated genes (81% of R5020% and 63% of E2) were not shared by both hormones (*Figure 1—figure supplement 2B*). Genes like *PGR* (progesterone receptor) and cell-cycle regulator *CCND2* (cyclin d2) were upregulated by E2 but not by R5020, while *TGFA* (transforming growth factor alfa) was upregulated by both hormones under these single-hormone treatments (*Figure 1—figure supplement 2B and C*).

Compared to T47D cells, PR protein levels in Ishikawa cells were significantly lower (*Figure 1—figure supplement 3A and B*). Copy number variation analysis revealed that Ishikawa cells carry neither chromosomal alterations nor multiple copies of the PGR locus as was shown in T47D cells, possibly accounting for differences in PR content between both cell lines (*Figure 1—figure supplement 3C*; *Le Dily et al., 2014*). To support the functional activity of PR in Ishikawa cells, a reporter gene downstream of the MMTV promoter containing several Pg response elements (PREs) (*Figure 1E*) was transiently transfected and the cells were exposed to either R5020 10 nM or vehicle (OH) for

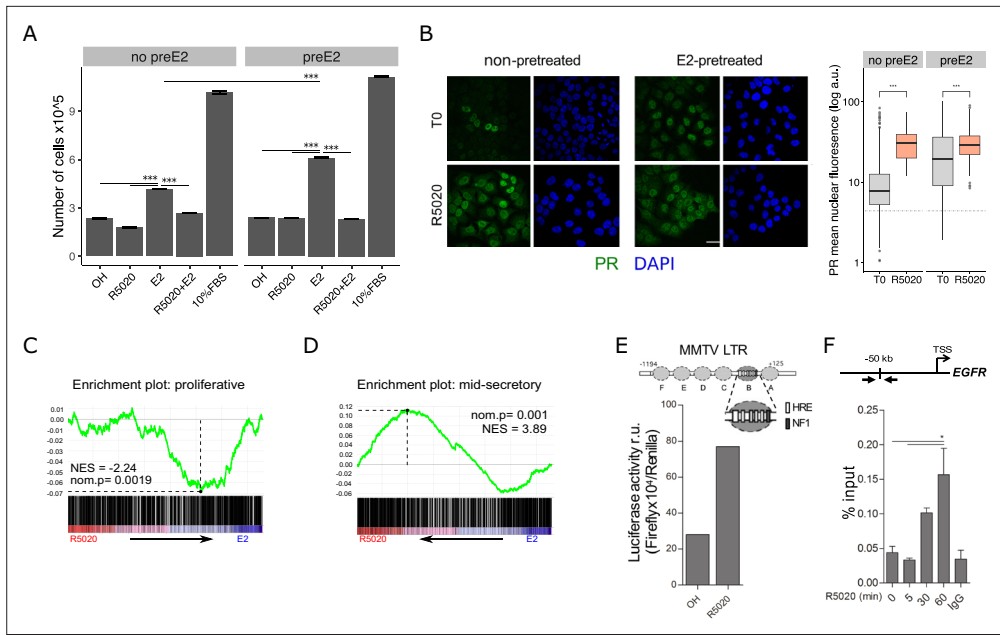

**Figure 1.** R5020 inhibits E2-induced Ishikawa cell proliferation through an active PR that is capable of transactivating an exogenous MMTV promoter sequence and an endogenous enhancer sequence located 50 kb upstream of EGFR gene. (**A**) Proliferation of Ishikawa cells either pretreated with E2 10 nM for 12 hr (preE2) or not (no preE2) and later treated with vehicle (OH), E2 $10\,nM$ (E2), R5020 $10\,nM$ (R5020), E2 combined with R5020 (E2+R5020) and FBS (10%FBS), expressed as mean number of cells ± SE of three independent experiments. (***) p < 0.001. (**B**) Immunofluorescence of PR in untreated (T0; top left), 60 min R5020-treated (R5020; bottom left), 12 h E2-pretreated (top right) and 12 hr E2-pretreated 60 min R5020-treated (bottom right) Ishikawa cells. Scale bar is equivalent to 30 m. Mean nuclear signal of PR for every cell in all images was determined and shown to the right of the images as arbitrary units (log a.u.). Horizontal dashed lines in boxplots indicate background signal for secondary antibody. (***) p < 0.001. (**C and D**) Gene set enrichment analysis (GSEA) results using R5020- and E2-treated Ishikawa expression profiles as phenotypes for classification of normal endometrium (proliferative and secretory) samples. Enrichment profile (green) shows correlation of normal samples at the top or bottom of a ranked gene list (phenotypes). Normalized enrichment scores (NES) and nominal p values (nom.p) are shown in the graphs. (**E**) Ishikawa cells transfected with an MMTV-Luciferase reporter gene and treated with vehicle (OH) and R5020 $10\,nM$ (R5020) for 18 hr. Diagram at the top depicts MMTV LTR promoter features, including several hormone response elements (HRE) and a nuclear factor 1 (NF1) binding site within nucleosome B (dark gray circle and magnification). Numbers in the diagram indicate base pair position relative to transcription start site (TSS). Results are expressed as relative units (r.u.) of Luciferase activity. (**F**) Representation of *EGFR* TSS and the enhancer sequence located 50 kb upstream used to evaluate PR recruitment. Black arrows indicate position of qPCR primers employed on samples treated or not (0) with R5020 for 5, 30, and 60 min. Unspecific immunoprecipitation of chromatin was performed in parallel with normal rabbit IgG (IgG). Results are expressed as %input DNA and bars represent mean fold change in PR enrichment relative to time 0 (untreated cells) ± SE of two independent experiments. (*) p < 0.05.

The online version of this article includes the following source data and figure supplement(s) for figure 1:

**Figure supplement 1.** E2-dependent cell cycle progression and proliferation of Ishikawa cells is mediated by ERalpha.

**Figure supplement 2.** RNAseq results from hormone-treated Ishikawa cells.

**Figure supplement 3.** Ishikawa cells express six times less PR than T47D cells.

**Source data 1.** Proliferation assays in treated and untreated Ishikawa cells.

**Source data 2.** Ishikawa mean nuclear signal intensity.

**Source data 3.** Ishikawa treated and untreated normalized expression dataset.

**Source data 4.** PR binding Ct values in *EGFR* enhancer sequence.

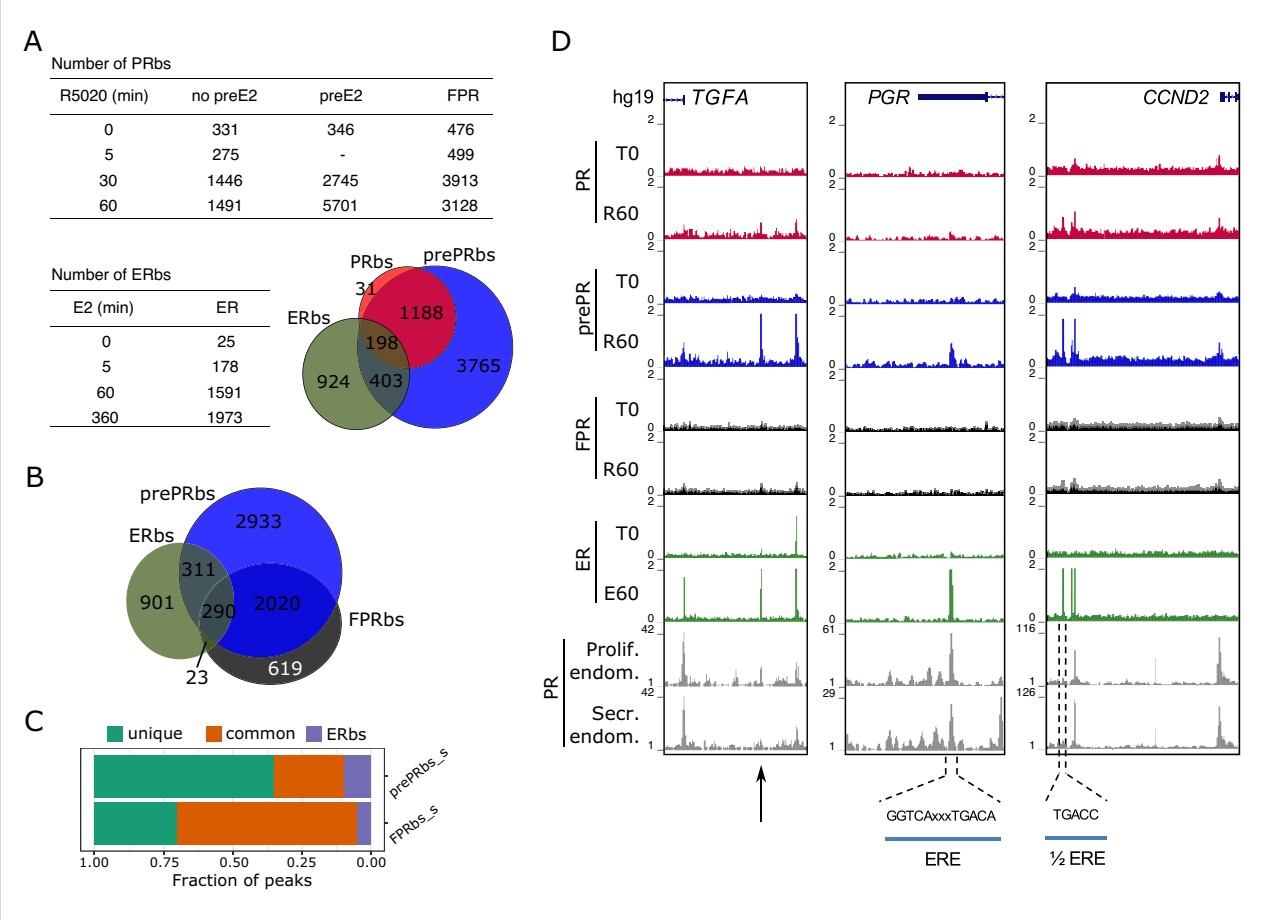

**Figure 2.** Estradiol induces R5020-dependent PR binding to specific regions in chromatin. (**A**) Upper table shows total number of PRbs obtained by ChIPseq for untreated (0 min) and R5020-treated (5, 30 and 60 min) endometrial Ishikawa cells under three different conditions: non-pretreated with E2 (PR), pretreated with E2 for 12 h (prePR) and exogenous expression of PR (FPR). Lower table shows number of ERbs using anti-ERalpha antibody on untreated (0 min) and E2-treated (5, 60 and 360 min) Ishikawa cells. Venn Diagram shows shared binding sites among PRbs (red), prePRbs (blue) and ERbs (green) at 60 min. (**B**) Venn Diagram shows intersection between ERbs (green), FPRbs (dark grey) and prePRbs (blue) at 60 min. (**C**) Fraction of peaks in FPR and prePR after substraction of shared PRbs (FPR_s and prePR_s, respectively) that are not shared with each other (unique), that are common to each other (common) and that are common with ER (ERbs). (**D**) Normalized coverage of PR and ERalpha binding in untreated (**T0**) and 60 min hormone-treated (R60 and E60) Ishikawa cells and PR binding in proliferative (GSE1327133) and secretory (GSE1327134) endometrium. Black arrow indicates peak of interest. R60: 60min R5020 $10\,\text{nM}$; E60: 60 min E2 $10\,\text{nM}$. The three regions displayed include *TGFA*, *PGR* and *CCND2* genes (indicated at the top). An estrogen response element (ERE) and a half ERE are indicated below the peaks.

The online version of this article includes the following figure supplement(s) for figure 2:

**Figure supplement 1.** Genome-wide analysis of PR and ERalpha binding in Ishikawa cells.

**Figure supplement 2.** PR overexpression on Ishikawa cells.

20 hours. A 2.8 fold increment in Luciferase activity was found in cells exposed to R5020 relative to OH controls, supporting the capacity of ligand-activated PR to activate PREs (*Figure 1E*). Binding of PR to cellular chromatin targets was confirmed by ChIP analysis of the PRE at the enhancer sequence located 50 kb upstream from the transcription start site (TSS) of the *EGFR* (epidermal growth factor receptor) gene (*Figure 1F*; *Figure 1—source data 4*). Together, these observations demonstrate that PR is responsive to progestins and functional in Ishikawa cells under our culture conditions and that these cells resemble E2 and Pg hormone regulation to proliferative and secretory normal endometrium respectively.

## Binding of PR and ERalpha to the ishikawa endometrial cancer genome

To explore the genome-wide distribution of PR and ERalpha binding (PRbs and ERbs respectively) in Ishikawa cells, ChIPseq was performed in different conditions (*Figure 2A* and *Figure 2—figure*

*supplement 1*). First, we analyzed untreated cells (T0) and cells exposed for 5, 30, and 60 min to 10 nM R5020 using a specific antibody to PR that detects both isoforms, PRA and PRB. Results showed robust PR binding after 30 min of R5020 treatment (R5020 30 min) with 1,446 sites, of which 322 sites (22%) were present in untreated cells (PRbs at time zero, T0 = 331). After 60 min of treatment with R5020 (R5020 60 min), the majority of sites identified at 30 min were still evident (78%) (*Figure 2A* and *Figure 2—figure supplement 1A*). The representation of PREs in 22% of the PR binding sites that were lost between 30 and 60 min of R5020 treatment was analyzed taking into account common, and unique 30 min or 60 min PRbs. De novo motif discovery, analysis of information content and quantification occurrences of PRE motifs in such regions did not show differences in the information content (the strength of PRE motif), nor new motif different from PRE, but revealed a higher abundance of PREs in common and unique 60 min datasets, yielding 1.72 fold and 1.78 relative unique sites in 30 min respectively. qPCR performed on six regions in the vicinity of hormone regulated genes and occupied by PR at 30 and 60 min of R5020 exposure validated ChIPseq results (*Figure 2—figure supplement 1B*). These regions were selected according to differentially expressed genes from RNAseq data and top-ranked by peak signal. These results indicate that hormone-dependent PR occupancy increased 5-fold by 30 min and stabilized between 30 and 60 min of treatment, in accordance with qPCR results (*Figure 2—figure supplement 1C*).

Next, we explored the recruitment of ERalpha to chromatin of Ishikawa cells exposed to E2 (10 nM) for 5, 60 and 360 min. Poor ERalpha binding was detected at T0 (25 sites), of which 90% remained occupied throughout all times of treatment with E2. Exposure to E2 resulted in the detection of 178 ERalpha binding sites (ERbs) at 5 min, 1591 at 60 min and 1,973 at 360 min (*Figure 2A* and *Figure 2—figure supplement 1D*). The majority (85%) of ERbs found at 60 min was also identified at 360 min (*Figure 2—figure supplement 1D*). ERalpha binding at 0, 60, and 360 min of E2 treatment was confirmed by qPCR on four of the sites identified (*Figure 2—figure supplement 1E*). ChIPseq results point to a clear and sustained E2-dependent enhancement of ER alpha binding (*Figure 2—figure supplement 1F*).

De novo motif discovery confirmed that PR binding occurred mostly through PREs exhibiting the complete palindromic response elements (*Figure 2—figure supplement 1G*), while ER binding sites were enriched in half-palindromic ERE motifs (*Figure 2—figure supplement 1H*). Comparison with previous findings in T47D cells (*Ballaré et al., 2013*) enabled clustering of both PRbs and ERbs into two classes (*Figure 2—figure supplement 1G* and H, respectively): sites specific for Ishikawa cells (group I; 595 PRbs, group III: 1,101 ERbs) and sites present in both Ishikawa and T47D cell lines (group II: 896 PRbs; group IV: 490 ERbs). Classification revealed that PR binds through complete PREs regardless of cell line identity, but in Ishikawa cells ERalpha binds mostly sites with only half of the characteristic palindrome.

## Estrogenic environment defines the landscape for PR binding to the endometrial genome

Shifts in the synthesis and secretion of the ovarian steroids (estrogen and progesterone) during the menstrual cycle serve as the principal hormonal drivers for endometrial changes. Rising circulating estradiol during the mid-to-late follicular phase of the cycle promotes the proliferation of the functional endometrium, and higher E2 levels upregulate *PGR* gene expression (*Graham et al., 1995*; *Kraus and Katzenellenbogen, 1993*). A similar result was reported in Ishikawa cells treated with E2 (*Diep et al., 2016*). To explore the effect of E2 on PR binding to DNA we performed PR ChIPseq analyses on Ishikawa cells exposed to E2 10 nM for 12 hr (preE2) before treatment with R5020 for 30 and 60 min. Pretreatment with E2 significantly increased the number of R5020-dependent PRbs (prePRbs), which included most of PRbs already identified in non-pretreated Ishikawa cells (*Figure 2A*, Table and Venn Diagram). Quantitative real-time PCR validations performed on six sites occupied by PR confirmed positioning of the receptor in both non-preE2 (non E2 pre-treatment) and preE2 conditions (*Figure 2—figure supplement 2A*). It also showed that E2 pretreatment augments both number of PRbs and occupancy of the receptor (signal) (*Figure 2—figure supplement 2B*). Contrary to PRbs in non-pretreated cells, the number of PRbs doubled between 30 and 60 min of R5020 in preE2 cells, reaching 5701 sites (*Figure 2A*). All conditions replicated their binding profiles in an independent experimental setting (*Figure 2—figure supplement 2C*).

Sequencing experiments performed on T47D cells exposed to 10 nM R5020 revealed over 25,000 PRbs (*Ballaré et al., 2013*; *Nacht et al., 2016*), likely reflecting the high content of PR in these cells. However, a large proportion of these PRbs was considered functionally irrelevant as indicated by the lack of nucleosome remodeling in response to hormone treatment (*Ballaré et al., 2013*). More recent experiments in T47D exposed to subnanomolar R5020 revealed that around 2000 PRbs are sufficient to evoke a functional response (*Zaurin et al., 2021*). Hence, the number of PRbs found in Ishikawa cells probably reflects the low concentration of PR, which is compatible with a functional response to progestins. To test this possibility we increased the levels of PR in Ishikawa cells by expressing a recombinant FLAG-PR vector. These cells, FPR Ishikawa (FPR), expressed levels of PR comparable to T47D cells (*Figure 2—figure supplement 2D*) and showed no impairment in hallmark phosphorylation of serine 294 in PR (*Figure 2—figure supplement 2E*), indicating that FPR cells were capable of responding to hormone. Upon hormone exposure, FPR cells exhibited rapid binding of PR to the *EGFR* enhancer sequence (*Figure 2—figure supplement 2F*). ChIPseq experiments after R5020 exposure showed twice the number of PRbs in FPR cells compared to parental Ishikawa cells. The majority of PRbs identified in Ishikawa cells ( > 90%) were also detected in FPR cells (*Figure 2—figure supplement 2G*), meaning that PR overexpression reflected mostly on an increase in number of binding sites.

Upon hormone induction, sites engaged by PR in Ishikawa cells were also occupied in FPR and pretreated cells, denoting a strong similarity between them. Although a small number of binding sites was shared between ERalpha and PR in all three conditions, PR binding in pretreated cells exhibited a higher degree of similarity to ERalpha binding than FPRbs (*Figure 2B*). Moreover, subtracting PRbs from FPRbs (FPR_s) and prePRbs (prePRs) heightens this difference, with a much larger fraction of binding sites shared with ERalpha in the case of prePRbs (*Figure 2C*). Among these sites, one located close to the promoter of *TGFA* gene, identified as an ERbs, showed significant PR binding only in preE2 Ishikawa cells, but not in FPR (*Figure 2D*, left panel). ERE-containing ERbs, such as the ones found in the transcription termination site of *PGR* gene and immediately upstream of *CCND2* promoter, were occupied by R5020-bound PR in preE2 Ishikawa cells (*Figure 2D*, middle and right panels). These three genes were upregulated by E2 treatment in RNAseq experiments performed on Ishikawa cells (*Figure 1—figure supplement 2C*). Importantly, PRbs identified in non-pretreated and E2-pretreated Ishikawa cells resembled the PR binding profile found in samples from normal proliferating and mid-secretory endometrium (*Chi et al., 2020*; *Figure 2—figure supplement 2H*).

The distribution of PRbs and ERbs in non-pretreated Ishikwa cells, in FPR cells and cells pretreated with E2 (prePRbs) relative to TSS of regulated genes was consistent with previous reports in oher cell lines (*Ballaré et al., 2013*; *Need et al., 2015*), in that they were enriched in intronic and distal intergenic regions (*Figure 3A*). Nearly 50% of binding sites localized to distal regions ( > 50 Kb) and approximately 30% to introns other than the first intron, indicating that regulation of gene expression by the steroid receptors PR and ERalpha is not mediated through proximal promoters but mostly by distal enhancer/silencer sequences. We corroborated these results employing another strategy based on binding site-gene association using the GREAT web tool (see Materials and methods for further details *McLean et al., 2010*). First, we defined a set of genes associated to binding sites with a basal plus extension rule (extended up to 100 kb away) and then we intersected this group of genes with R5020- or E2-regulated genes. Of the 1886 genes regulated by R5020, only 224 (12%) were potentially associated to PRbs, while only 199 of the 950 genes regulated by E2 (21%) proved to be associated to ERbs (*Figure 3B*).

As expected, from the sequences contained in 10 kb windows centered in peak summits of PRbs, FPRbs and prePRbs, the PRE emerged as the most representative binding motif (*Figure 3C*), including sites uniquely found in FPR (group c: 633) or preE2 (group d: 3,162) cells. While comparison between ERalpha and PR ChIPseq results showed few similarities regarding identity of binding sites, with a set of 216 shared by both hormone receptors, pretreatment with E2 added nearly twice as many binding sites to the pool shared with ERalpha (from *Figure 2A*, Venn Diagram). The most representative motif discovered in these sites -only shared by ER alpha and prePR- was a half ERE (*Figure 3D*, group h: 329) that was highly similar to the motif observed in sites uniquely found in Ishikawa ERbs (from *Figure 2—figure supplement 1H*, group III). Sites shared by ERalpha and PR in all three conditions resulted in an unclear combination of PRE and ERE motifs (*Figure 3D*, group e-g). Degenerated motif logo in group g showed no association to any known motif, probably due to a corrupt analysis performed on

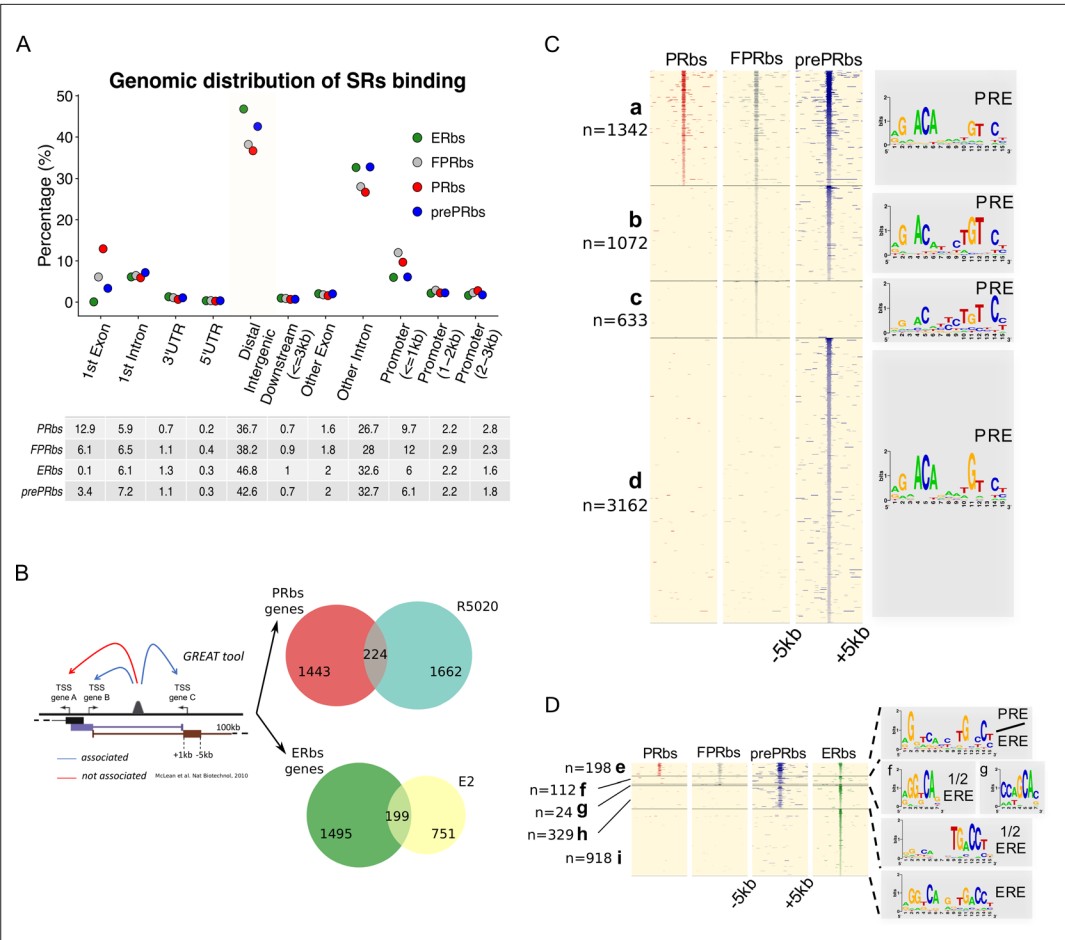

**Figure 3.** A fraction of E2-induced PRbs localize on ERbs and contain half ERE motifs. (**A**) Classification of steroid receptor binding relative to genomic features expressed as percentage (%) of peaks after 60 min of hormone treatment inside each feature. Legend at the top right corner indicates the color key for ERbs (green dots) and three conditions of PR binding: non-pretreated with E2 (PRbs, red dots), pretreated with E2 for 12 hr (prePRbs, blue dots) and exogenous expression of PR (FPRbs, grey dots). The table below shows percentages represented in the plot. (**B**) To the left: Representation of GREAT tool association rules adapted with modifications (*McLean et al., 2010*). To the right: Venn diagrams show intersection between PRbs-associated genes and R5020-regulated genes (top), and ERbs-associated genes and E2-regulated genes (bottom). (**C**) Peak signals in PRbs, FPRbs and prePRbs from 60 min R5020-treated Ishikawa cells were plotted as heatmaps. Regions were defined inside a window of 10 kb centered in peak summit (±5 kb) and intensity of the signal correspond to number of reads in each region. Heatmap is subdivided into four mutually exclusive groups depending on shared/partly shared/non-shared binding sites: a (n = 1342), sites shared by all three conditions of PR binding; b (n = 1072), sites uniquely found in FPR and prePR; c (n = 633), sites found only in FPR; and d (n = 3162), sites found only in prePR. De novo motif discovery (MEME) was performed on all groups and results are indicated as sequence logos to the right of the map, including the name of the most related known motif. PRE: progesterone response element. (**D**) Peak signals in PRbs, FPRbs and prePRbs as in (**C**), and ERbs from 60 min E2-treated Ishikawa cells. Heatmap was subdivided into five mutually exclusive groups: e (n = 198), sites shared by all three conditions of PR binding and ER binding; f (n = 112), sites shared by FPRbs, prePRbs and ERbs; g (n = 24), sites shared by FPRbs and ERbs; h (n = 329), sites shared by prePRbs and ERbs; and i (n = 918), sites uniquely found in ERbs. Motif discovery was performed as in A for all groups and results are shown to the right of the map, including the most related known motif. ERE: estrogen response element; 1/2 ERE: half ERE.

insufficient data, and the partially degenerated motif logo in group e showed limited association to both PRE and ERE (PRE/ERE).

Taken together, this evidence suggests that, provided there is an estrogenic background, activated PR could regulate estrogen-dependent Ishikawa-specific transcriptome by binding sites already or formerly bound by ERalpha.

# PAX2 binds chromatin in close proximity to ER alpha and PR binding sites in Ishikawa cells

Evidence described so far partially explains cell type specific hormone-dependent gene regulation, although it is not sufficient to understand the mechanisms underlying differential binding of hormone receptors to chromatin. Initially, we addressed this by contrasting the sequences of ERbs and PRbs from groups I-IV, that is hormone regulated Ishikawa specific (from *Figure 2—figure supplement 1G and H*) with an array of 1395 known TF binding motifs (see Materials and methods). Results revealed an enrichment (p-value < 1 e-4) of multiple members of the PAX family -including variants 2, 5, 6, and 9- in groups I and III, that is In Ishikawa-specific PRbs and ERbs (*Figure 4—figure supplement 1A* and B, respectively), suggesting that members of the Pax family may be involved in PR and ER alpha action in Ishikawa cells. Unbiased comparison (all sites) of enrichment in TF binding motifs between Ishikawa and T47D PRbs showed similar results for PRbs, although the enrichment was less significant (*Figure 4—figure supplement 1C*). Moreover, while enrichment of PAX motifs was also observed around ERbs in Ishikawa cells (*Figure 4—figure supplement 1D*), this was not the case with T47D cells, in which examples like the well-known breast-related pioneer transcription factor FOXA1, were found instead (*Figure 4—figure supplement 1E*).

Enrichment of NR3C1-4 (mineralocorticoid, glucocorticoid, progesterone, and androgen receptors) and ESR1 motifs included into the 1395 known motifs corroborated de novo discovery performed with MEME in both Ishikawa and T47D cells. Stronger enrichment of PAX motifs was observed in prePRbs compared to PRbs (*Figure 4A*), indicating that PR binding to regions potentially bound by PAX is favored after E2 pretreatment. Coherently, while equivalent fold enrichment values were detected when comparing prePRbs to ERbs (*Figure 4B*), comparison between prePRbs and FPRbs showed that increased PR levels alone were not sufficient for a greater association to PAX binding motifs (*Figure 4—figure supplement 1F*). Consistently, RNAseq experiments on Ishikawa cells treated either with R5020 10 nM or E2 10 nM for 12 hr showed putative PAX2 binding sites among the top 20 significantly enriched TFs (DAVID web-based tool *Huang et al., 2009*) on differentially regulated genes (*Figure 4C*). PAX2 and the ubiquitous AP1 were the only TFs (including other PAX family members) predicted to bind both E2 and R5020-regulated genes. ER was also predicted to bind on E2-responsive genes, while glucocorticoid receptor (GR) motif (PR-like motif) was detected on R5020-responsive genes.

Next, we evaluated the expression of *PAX2* and other family members in Ishikawa RNAseq samples (*Figure 5—figure supplement 1A*). Although none of the detected *PAX* genes seemed to be strongly regulated by R5020 or E2, expression of *PAX2*, *PAX3*, and *PAX5* was found to be lower and more variable (comparing experimental replicates) than that of *PAX6*, *PAX8*, and *PAX9*. Despite expression levels of the latter were markedly higher, robust evidences indicate that reduced levels or absence of PAX2 were strongly associated to early onset of neoplastic processes in endometrial tissue and precancerous lesions (*Monte et al., 2010*; *Raffone et al., 2019*). We corroborated those findings using publicly available data comprising of several RNAseq samples from normal and cancerous endometrial tissue (*Dou et al., 2020*), in which expression of *PAX2* was significantly reduced in tumor samples (Endometrioid and Serous adenocarcinomas) compared to normal tissue (*Figure 5A*).

PAX2 association to PR and ER alpha action was also evaluated by immunofluorescence against PAX2. Nuclear localization of PAX2 was observed predominantly after 60 min of R5020 in pretreated and non-pretreated PR+ cells (*Figure 5B*), indicating that hormonal treatment promotes co-localization of PAX2 and PR in nuclei of Ishikawa cells. Similar results in PAX2 localization were obtained after treating Ishikawa cells with E2 for 60 min (*Figure 5—figure supplement 1B*). The increase in nuclear PAX2 signal is not due to changes in protein levels, which were not affected by treatment with either R5020 or E2 (*Figure 5—figure supplement 1D*). In accordance to motif analysis results, PAX2 was not detected in nuclei of T47D cells after hormonal treatments (*Figure 5—figure supplement 1D*).

To extend these findings, we performed PAX2 ChIPseq experiments on untreated cells and in cells exposed for 60 min to either R5020 or E2. The results confirmed PAX2 binding to chromatin following hormonal treatment (*Figure 5C*). Even though identified PAXbs were few (T0: 43, R60: 202 and E60: 209), most of PAX2 binding occurred after R5020 and E2 treatments. Moreover, PAX2 binding was not stochastically distributed in the genome of Ishikawa cells but rather partially associated to ERbs and PRbs (*Figure 5D*). This association was stronger for PR binding in cells pre-teated with E2 than in non-pretreated cells or in cells overexpressing recombinant PR (*Figure 5E and F*). Similar results were

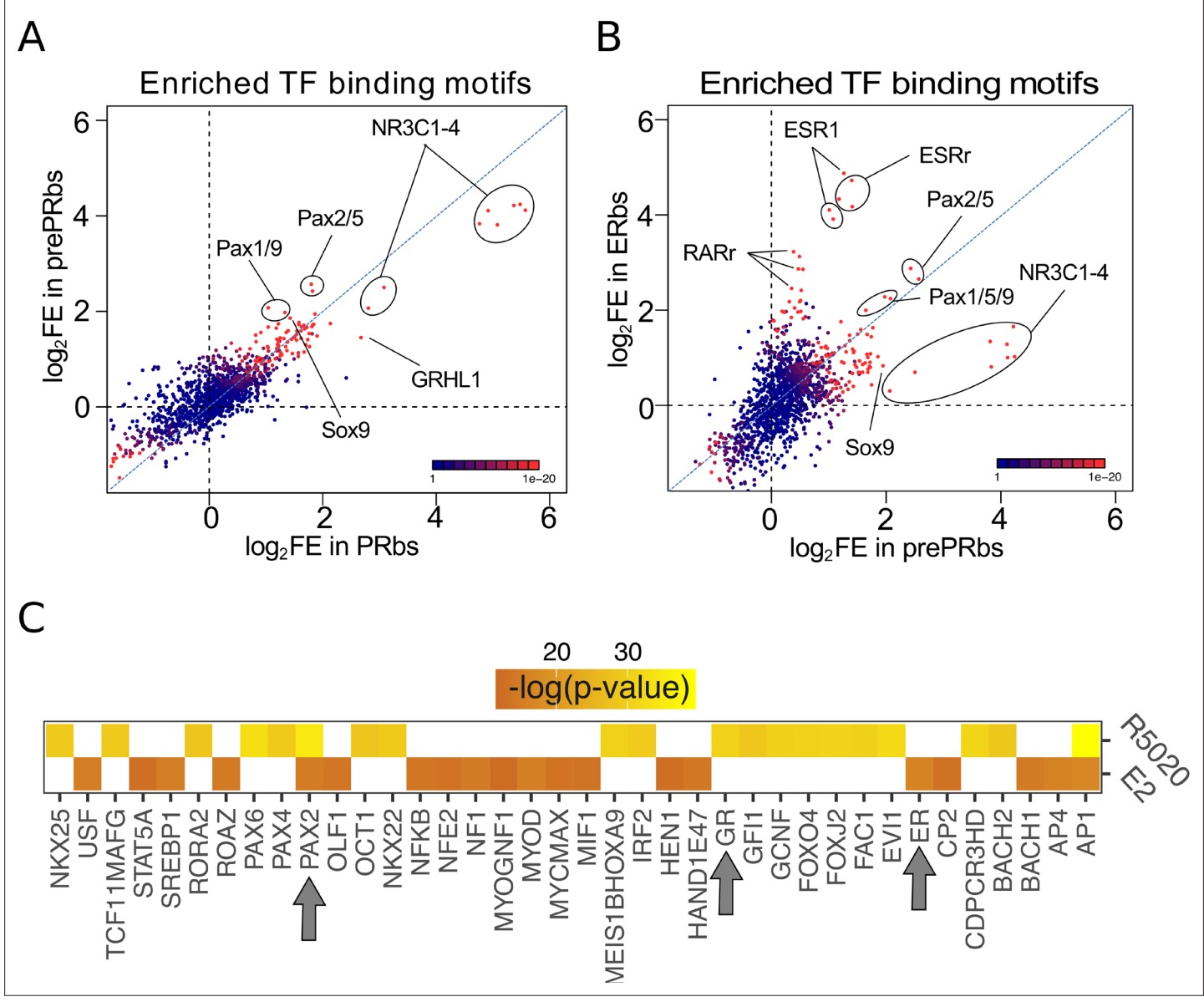

**Figure 4.** Putative PAX2 binding sites are associated with PR and ER alpha binding and hormone-regulated genes in Ishikawa cells. (**A**) Fold enrichment values (log2FE) of 1.395 known TF binding motifs on prePRbs and PRbs. Combined p-values for enrichment analyses are indicated through the color key displayed at the lower right corner of the plot. Relevant motifs pointed on the plot correspond to NR3C1-4, members of the PAX family (1, 2, 5, and 9) and SOX9. (**B**) Comparison as in (**A**) between prePRbs and ERbs. Relevant motifs pointed on the plot correspond to NR3C1-4, members of the PAX family (1, 2, 5, and 9), SOX9, ESR1 and estrogen related (ESRr) and retinoic acid receptor (RARr). (**C**) Predicted UCSC Transcription Factor (TFBS) binding on genes regulated by 12 hr treatments with R5020 10 nM and E2 10 nM in Ishikawa cells were analyzed using DAVID web-based functional enrichment tool. Heatmap shows the top 20 TFBS predicted (p < 0.05)for R5020- and E2-regulated genes from RNAseq results expressed as -log(p-value). Arrows indicate position of PAX2, GR (PR-like binding motif), and ER.

The online version of this article includes the following figure supplement(s) for figure 4:

**Figure supplement 1.** PAX2 binding to chromatin is hormone dependent in Ishikawa cells while it is not localized to nuclei of T47D cells.

observed for ERbs in response to E2 (*Figure 5E*). In agreement with the coverage plots, the association between PAX2 and binding of the receptors was statistically significant (two-tailed Fisher's exact test; p-value < 0.001). On the contrary, we found no significant overlap between randomly rearranged PAXbs (shuffled PAXbs) and binding of receptors (two-tailed Fisher's exact test; p-value > 0.1). These results indicate that PAX2, and possibly other members of the PAX family may co-operate with PR and

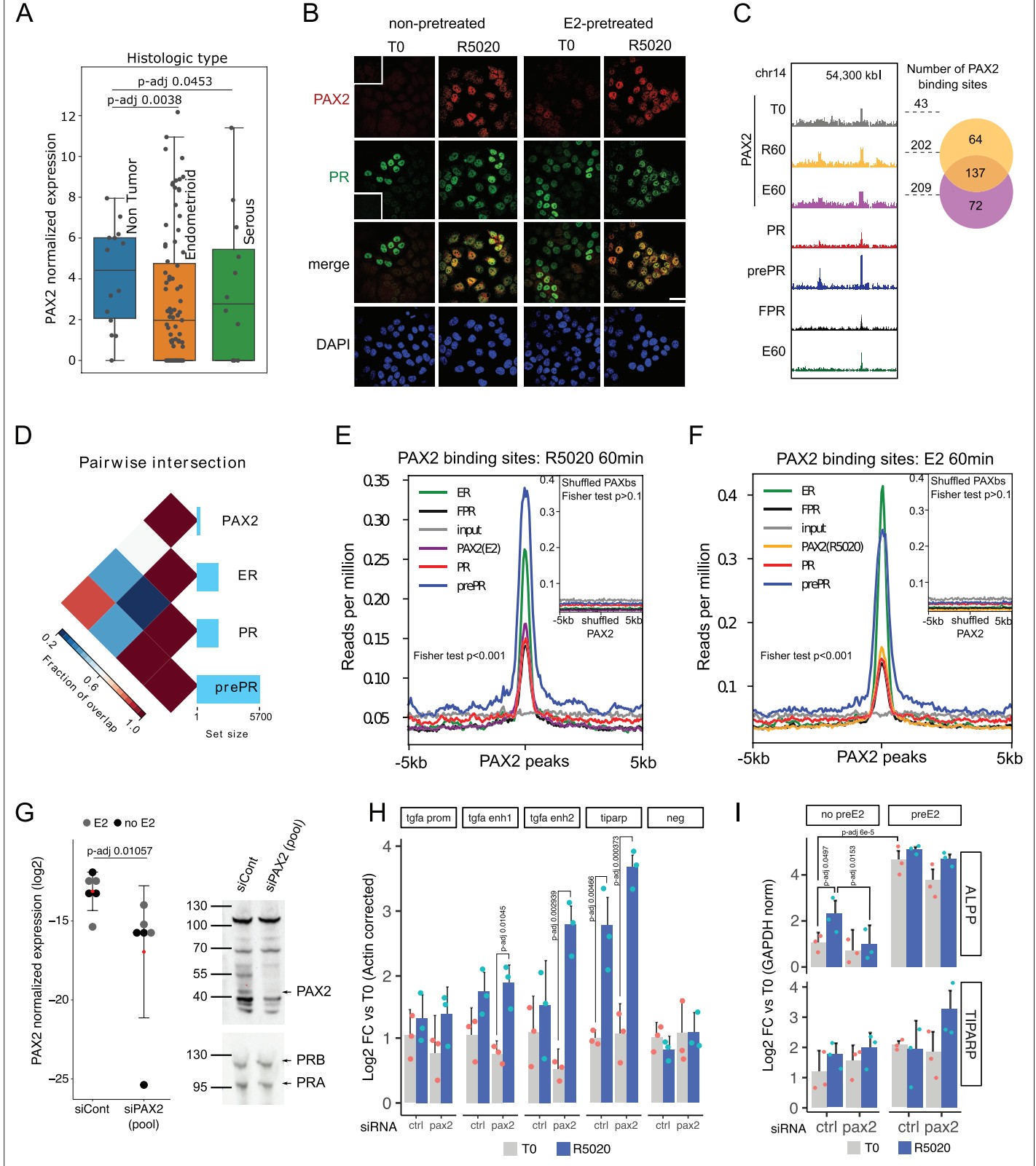

**Figure 5.** PAX2 co-localizes with PR and ERalpha in the nuclei of Ishikawa cells and it is positioned in the vicinity of PR and ER alpha binding sites. (**A**) PAX2 expression in Non Tumor (normal endometrium) and two histologically distinct endometrial cancer samples (***Dou et al., 2020***). Data stored at the National Cancer Institute's CPTAC program was accessed through cptac python package developed by Sam Payne lab. (**B**) Immunofluorescent detection of PR (green) and PAX2 (red) in untreated (T0) and 60min R5020-treated (R5020) Ishikawa cells which were pretreated or not with E2 for 12h

*Figure 5 continued on next page*

*Figure 5 continued*

(non-pretreated, E2-pretreated). Images were merged for co-localization analysis (merge). Scale bar is shown in the panels and is equivalent to 30 µm. (**C**) PAX2 binding profile inside a region of 70kb of chromosome 14. Number of PAX2 binding sites for untreated Ishikawa cells and treated with R5020 for 60 min or E2 for 60 min is shown to the right of the profiles as well as the intersection between these last two groups (venn diagram). Tracks for PRbs, prePRbs, FPRbs and ERbs are displayed below the profiles for the same region. (**D**) Pairwise intersections of the fraction of overlap between PRbs, prePRbs, ERbs and PAXbs. Color key is indicated below the plot and the size of each set is shown as a bar plot to the right side. Intervene software was employed in the analysis (*Khan and Mathelier, 2017*). (**E**) Binding profiles of ER (green), PR (red), FPR (black), and prePR (blue) on PAX2 binding sites of 60min R5020-treated Ishikawa cells. PAX2 binding after 60min E2 treatment was included (purple). Inset shows signal profiles centered on shuffled R5020-dependent PAX2 binding sites. P-value for fisher extact test is reported in the plots. (**F**) Binding profiles as in (**E**) on PAX2 binding sites of 60min E2-treated Ishikawa cells. PAX2 binding after 60min R5020 treatment was included (orange). As in (**E**), inset shows signal profiles centered on shuffled E2-dependent PAX2 binding sites. p-Value for fisher extact test is reported in the plots. (**G**) PAX2 knockdown evaluated by qPCR (left panel) and Western blot (right panel) after treating Ishikawa cells with a control siRNA (siCont) or a mix of specific siRNAs (pool). PCR data from three replicates were normalized by *GAPDH* expression and shown in the plot together with the mean± SEM. PR was used as control for western blot. (**H**) Binding of PR in cells treated with siRNA control or siPAX2 followed by 12 hr pretreatment with E2 and treatment with R5020 for 60min. PCR signal was corrected by beta-actin and relativized to untreated control cells. Bars show the mean± SEM of the three replicates displayed as dots. (**I**) Gene expression levels of hormone responsive genes before and after siPAX2 treatment. Expression was normalized to *GAPDH* and relativized to T0 using ΔΔCt method and expressed as mean log2 fold change± SEM of the three replicates displayed as dots. For all items, statistically significant comparisons resulting from ANOVA followed by Tukey's HSD are denoted by the adjusted p-value.

The online version of this article includes the following source data and figure supplement(s) for figure 5:

**Source data 1.** PR recruitment measured by qPCR in PAX2-knocked down Ishikawa cells.

**Source data 2.** Gene expression measured by qPCR in PAX2-knocked down Ishikawa cells.

**Figure supplement 1.** Expression analysis of PAX2 and other PAX family members in Ishikawa cells.

**Figure supplement 2.** Expression analysis of hormone regulated genes after PAX2 knock down.

ERalpha for binding to chromatin in Ishikawa cells but not in T47D cells, in which neither enrichment for PAX binding motif nor nuclear localization of PAX2 was detected.

In line with evidence on the role of *PAX2* in endometrial cancer, we studied the effect of down-regulating expression levels of *PAX2* in Ishikawa cells using a mix of specific siRNAs that produced a marked reduction of its transcript and protein (*Figure 5G*). Consistently with results described above, pretreatment with E2 for 12 hr did not alter levels of PAX2 or interfered with its knock down. Interestingly, knock down of *PAX2* under estrogenic conditions led to a significant R5020-dependent increased binding of PR to regulatory regions upstream of *TGFA* and in *TIPARP* promoter (*Figure 5H*, *Figure 5—source data 1*), which suggest that lack of PAX2 is probably exacerbating E2 stimulation on PR recruitment. No increment in PR binding was detected in the promoter of TGFA or in an unrelated genomic region (negative).

This effect did not seem to alter the expression of hormone responsive genes in a similar fashion. In the absence of E2 pretreatment, R5020-dependent expression of *ALPP* was abolished by PAX2 knock down, while *TIPARP* regulation was practically unaffected (*Figure 5I*, *Figure 5—source data 2*). This is consistent with the fact that *TIPARP*, but not *ALPP*, had its promoter regulatory region occupied by both PR and ERalpha upon hormone stimulation (*Figure 5—figure supplement 2A*). Thus suggesting that PAX2 could modulate PR binding in regions potentially co-occupied by ER alpha. This effect was corroborated under estrogenic conditions and although reduced levels of PAX2 clearly affected the gene regulatory profiles of both genes, differences were not statistically significant probably due to the variable gene expression background in Ishikawa cells and residual levels of PAX2 among replicates. As was observed in the control condition (non-target siRNA), E2-pretreatment increased *ALPP* transcript levels after PAX2 knock down (pretreated vs. non-pretreated cells), which were then further increased by R5020 treatment. However, as in non-pretreated cells, *ALPP* transcript levels were lower compared to siRNA control conditions. In the case of *TIPARP*, we found that in agreement with previous reports on breast cancer cells (*Rasmussen et al., 2021*), expression levels were increased by E2 (pretreated vs. non-pretreated cells) and apparently by R5020 as well (treated vs. untreated conditions). In line with our hypothesis, supported by PR recruitment to *TIPARP* promoter region, R5020 treatment increased *TIPARP* levels in PAX2-knocked down cells.

These results suggest that the lack of PAX2 resembles some characteristics of an unopposed estrogen action, promoting reduced levels of *ALPP* and increased levels of *TIPARP*. Moreover, PAX2 knockdown reduced *ESR1* and *PGR* expression (*Figure 5—figure supplement 2B*) pointing to a

pivotal role of PAX2 in sustaining PR and ERalpha target gene regulation. However, it did not affect *TGFA* levels (*Figure 5—figure supplement 2B*) that were substantially upregulated by E2 compared to non-pretreated cells, but downregulated by R5020 under estrogenic conditions.

## Under estrogenic conditions, PR and PAX2 conform endometrial regulatory domains in open chromatin compartments

Nuclear architecture is a major determinant of hormonal gene regulatory patterns (*Le Dily et al., 2014*). Therefore, we used in nucleo Hi-C technology to study the folding of chromatin across the genome of Ishikawa cells by generating genome-wide contact datasets of cells untreated (T0) or pretreated with E2 for 12 hr, and exposed to R5020 or E2 for 60 min. A comparison of contact matrices at 20 kb resolution of untreated Ishikawa cells to T47D cells confirmed the high degree of conservation on the borders of topologically associating domains (TADs) (*Figure 6—figure supplement 1A*). TADs are grouped into two chromatin compartments A and B, which represent the active open chromatin (A) and the closed inactive chromatin (B), respectively. Analysis of such compartments showed a cell type-specific patterning (*Figure 6—figure supplement 1B*), in which Ishikawa samples from two independent experiments were more closely related to each other than any of them to a T47D sample (*Figure 6—figure supplement 1C and D*). However, A/B profile distribution in Ishikawa cells was independent from hormonal treatments (*Figure 6—figure supplement 1B and E*), meaning that chromatin was in a primed state that conditioned hormone-dependent regulation of gene expression. Detailed analysis revealed that 7% of A domains in Ishikawa cells were B in T47D cells, and 12% of B domains in Ishikawa cells were A in T47D cells (*Figure 6—figure supplement 1F*). A total of 861 genes encompassed in the A compartment in Ishikawa cells belong in the B compartment in T47D cells, and 1438 genes in B compartments in Ishikawa cells belong in A in T47D cells (12%), suggesting that distribution of A and B compartments could in part explain cell type specific gene expression profiles.

To evaluate whether chromatin states are related to gene expression through differential binding of hormone receptors to DNA, we intersected PR and ER alpha ChIPseq results with the A/B compartment coordinates. Both transcription factors, PR and ER alpha, bound A compartments more frequently than B, meaning that open genomic regions in Ishikawa showed preferential binding of the hormone receptors (*Figure 6—figure supplement 1G*). Neither pre-treatment with E2 nor expression of recombinant PR modified the preferential binding of PR to the A compartments.

As mentioned above, PAX2 binding occurs mostly in close proximity to PR and ER alpha binding sites. In fact, distances between PAXbs and PRbs were remarkably shorter in E2 pretreated cells than in any other condition (*Figure 6—figure supplement 1H*). This raised the question of whether recruitment of PR together with PAX2 to open chromatin compartments facilitates regulation of gene expression. To study this notion, we defined putative endometrial regulatory domains that we named 'Progestin Control Regions' (PgCR) with the capacity to potentially regulate nearby genes. The restrictions for being a regulatory domain, which consisted in containing at least two PRbs separated by a maximun distance of 25 kb and a PAXbs (represented in *Figure 6A*: PgCRs Definition), were met mostly under E2 pretreated conditions. This outcome was due to the strong association between prePRbs and PAXbs, though it may have been aided by the increased PR protein levels. However, the sole increment in PR protein levels was not enough to force an association to PAXbs, given that FPR cells did not show similar results (*Figure 6A*: PgCRs Definition).

Considering that TAD borders may act as regulatory barriers (*Figure 6—source data 1*), we removed from further analysis any region that, in spite of satisfying the rules for being a PgCR, was localized across a barrier as well. In agreement with this restriction, the sizes of PgCR -with an average of 25kb- were smaller than TADs -with an average of 1000kb- (*Figure 6—figure supplement 1I*). In addition, the majority of the 121 identified PgCRs (coordinates in hg38 *Figure 6—source data 2*) were not located near the TAD borders, but in the TAD center (*Figure 6—figure supplement 1J*), where most non-housekeeping genes are found (*Le Dily et al., 2019*). Moreover, PgCRs seem to be located in A compartments in the vicinity of hormone-regulated genes like *PGR* and *ALPP* (*Figure 6A*). Regulation in the expression of these genes was validated by RNAseq and qPCR of total RNA samples of Ishikawa cells exposed to R5020, E2 and E2 followed by R5020 for 12 hr, which showed that *ALPP* is induced by both hormones and *PGR* is regulated by R5020 only after treatment with E2 (*Figure 5I*, *Figure 5—figure supplement 2B* and *Figure 6—figure supplement 1K*).

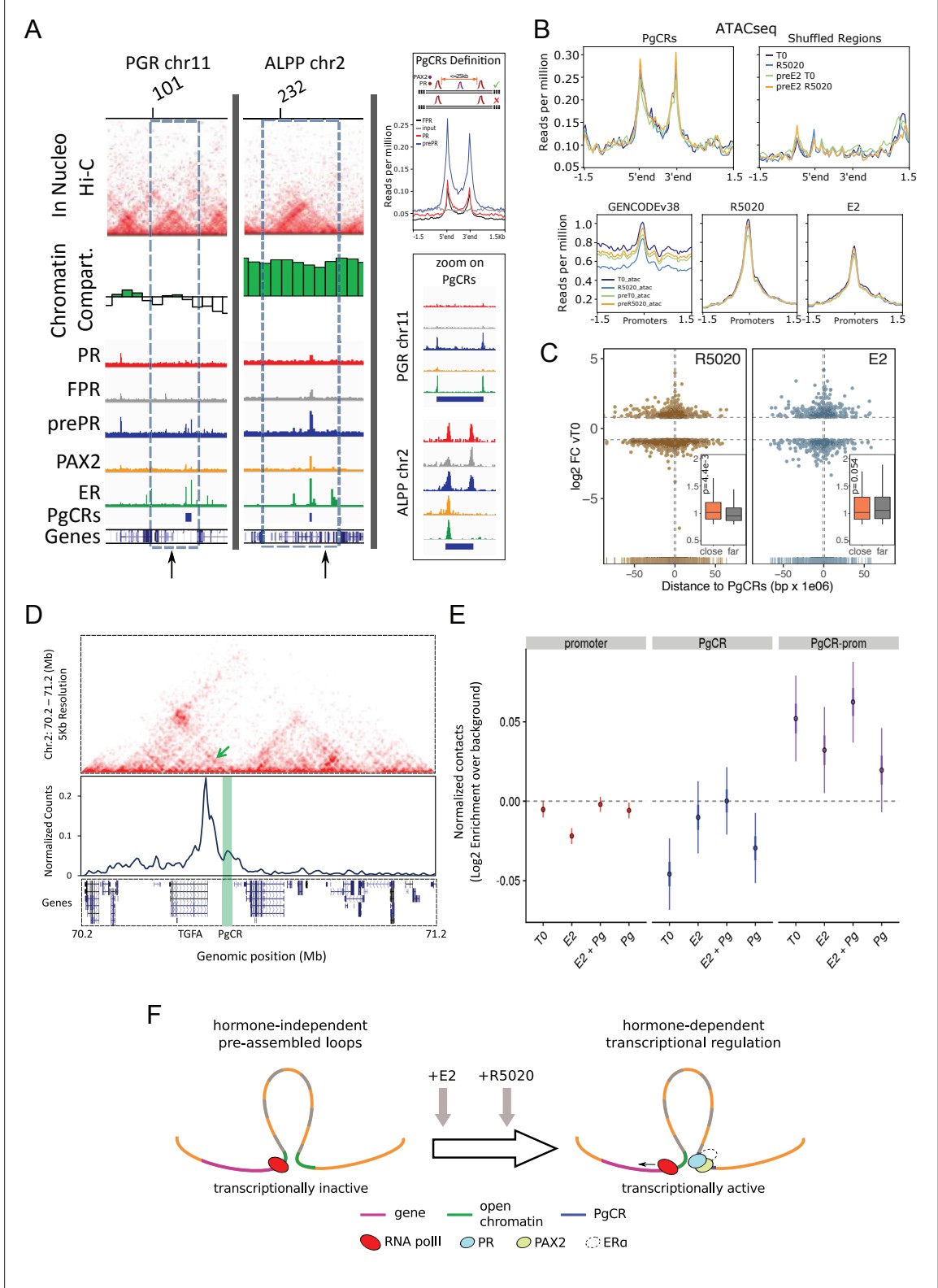

**Figure 6.** Convergence of PR and PAX2 binding in TADs with regulated genes defines potential endometrial regulatory domains. (**A**) Upper panel shows the contact matrices at a resolution of 20kb obtained by In Nucleo Hi-C in *PGR* and *ALPP* loci. Middle panel shows the spatial segregation of chromatin as open or closed compartments inside TADs (green bars: A compartment; white bars: B compartment - see Materials and methods section). The bottom panels show ChIPseq signal distribution of PR, FPR, prePR, PAX2 and ERalpha as well as the location of PgCRs and genes over the region.

*Figure 6 continued on next page*

*Figure 6 continued*

The dashed rectangle restricts the TAD of interest and the vertical arrow marks the TSS of *PGR* and *ALPP*. Definition of PgCR: Coverage profiles of PR (red), FPR (black), and prePR (blue) binding on Progesterone Control Regions (PgCRs) delimited by the start and end labels, and flanked upstream and downstream by 1.5Kb regions. Input sample (gray) was included in the plot. Rules for qualifying as a control region are depicted on top of the profile plot. Magnified images over Control Regions are shown to the right (zoom on PgCRs). (**B**) ATACseq peaks from cells untreated (T0), treated with R5020 for 60 min, 12hr E2-pretreated (preE2 T0) and E2-pretreated followed by 60min treatment with R5020. Signal was plotted over Control Regions, shuffled Control Regions (Shuffled Regions), promoters of all annotated genes from GENCODE database (GENCODEv38) and promoters of genes regulated by 12h treatments with R5020 or E2. (**C**) Plot shows fold change values of genes regulated by R5020 and E2 (v. untreated cells) relative to Control Regions. Genes located upstream of PgCRs are represented with negative distance values. Dashed horizontal lines mark fold change cut-off points (|log2FC|=0.8) and vertical lines are placed at position –1 and 1Mb. Insets depict comparison of fold change values (absolute values) between genes located beneath (close) and over (far) a 1Mb distance from PgCRs. Statistical significance for this comparison was determined with Welch Two Sample t-test and is represented by a p value on the plot. (**D**) Top panel: Hi-C contact map at 5kb resolution of chr2 (70,200,000-71,200,000) obtained in Ishikawa cells and showing the organization around TGFA gene locus. Middle panel: Virtual 4C profile at 5kb resolution (expressed as normalized counts per thousands within the region depicted above) using the TGFA promoter as bait and showing the contacts engaged between TGFA promoter and the PgCR detected in this region (highlighted in green). Arrow on top panel highlights the position of the loop in the map. Bottom panel shows the positions of genes in the region depicted. (**E**) Distributions of observed versus expected interactions established between promoters (red - left), between PgCRs (blue - middle) and between Promoters and PgCRs (purple - right) located within a same TAD in Ishikawa cells treated as indicated below. (**F**) Representation of a chromatin loop involving a PgCR and the promoter of a regulated gene. Initially, the gene is transcriptionally inactive even though the loop is already formed. After hormone induction (E2 pretreatment followed by R5020), PR, PAX2 and in some cases ER alpha occupy open chromatin compartments in contact with promoters resulting in transcriptional activation.

The online version of this article includes the following source data and figure supplement(s) for figure 6:

**Source data 1.** TAD coordinates (hg38).

**Source data 2.** PgCR coordinates (hg38).

**Figure supplement 1.** Chromosome compartments analysis and definition of progesterone control regions.

As was mentioned before, the Hi-C matrices were used to determine the spatial segregation of chromatin in both open and closed chromatin compartments (A/B), and the A:B ratio was independent of hormone treatment. Consistent with these results, ATACseq signal on PgCRs remained unchanged upon hormone exposure (*Figure 6B*, top panels). We evaluated the possibility that ATAC-seq signal on PgCRs was simply reproducing any potential region on the genome by relocating the reference regions ('shuffled regions') preserving their size and the chromosome they were originally in. Coverage signal was lost upon shuffling the reference regions indicating that chromatin was readily and non-randomly accessible to TFs in these locations. Although ATACseq peaks were also detected on promoters of hormone-regulated genes, the signal did not differ after hormone exposure (*Figure 6B*, bottom panels), implying that treatments were not responsible for opening the chromatin in these regions. In addition, both R5020- and E2-regulated genes with highest FC values (v. T0) were concentrated under 1 Mb ('close') away from PgCRs (*Figure 6C*), though the comparison between FC values of 'close' and 'far' (over 1 Mb) regulated genes was significant only in the case of R5020 (p=$4.4 \times 10^{-3}$; *Figure 6C*, inset).

Further analysis on Hi-C contact matrices revealed that PgCRs preferentially interact with promoters of hormone-regulated genes (*Figure 6D*). Employing a 'one-to-one' virtual 4 C approach we found that *TGFA* promoter was in close contact with a nearby PgCR (enhancer sequences validated by qPCR in *Figure 2—figure supplement 2A* and *Figure 5H*) possibly regulating the cyclical E2-dependent increment of the transcript and the R5020-dependent downregulation that followed. Although PgCR-promoter interactions were non-random and mostly intra-TAD, we found no difference in contact enrichment between treated and untreated cells (*Figure 6E*). These results are consistent with ATACseq profiles and imply that chromatin would be pre-assembled into regulatory loops–involving PgCRs and promoters–which are transcriptionally inactive until hormone-dependent binding of steroid receptors and PAX2 triggers PolII activation (*Figure 6F*).

These results suggest that specific binding of PR, PAX2, and ERalpha to chromatin occurs in compartments that are present in a permissive (open) or restrictive (closed) status depending on the cell line, and are not modified by short term hormone exposure (*Figure 6F*). However, it is not yet clear the role of PAX2 in PR binding to PgCRs. Summing up, PR and ER bind mostly to non-common sites that exhibit the corresponding consensus sequences, and are adjacent to PAX2 binding. Therefore, the endometrial specific hormone response results in part from specific chromatin compartments, unique receptor binding sites and selective TFs binding partners to regulate gene expression.

## Genes contained in TADs with PgCRs are associated to endometrial tumor progression

To explore the possibility that alterations in the expression profile of genes under the influence of PgCRs were related to disease progression such as endometrial cancer, we examined 497 RNAseq samples from a cohort of patients diagnosed with endometrial cancer (The Cancer Genome Atlas, TCGA, Project TCGA-UCEC). Processing and exploratory analysis of raw count data are described in Materials and methods section (*Figure 7—figure supplement 1A*). Samples were classified according to available clinical metadata, including the FIGO system (International Federation of Gynecology and Obstetrics), resulting in 488 samples of which 267 are Stage I, 53 are Stage II, 137 are Stage III and 31 are Stage IV. Principal component analysis (PCA) on these samples revealed a clear bias in the scattered distribution, mostly explained by the first component (PC1: 18%). Tumors in advanced stages (Stage II to IV) showed a tendency to group together and they were characterized by a marked reduction in *PGR* and *ESR1* levels (*Figure 7A*). However, just as it was observed for hormone-treated Ishikawa cells, no change in *PAX2* expression was detected throughout stages (*Figure 7—figure supplement 1B*). Expression profiles of selected genes previously implicated in tumor progression (causally or not) (*Liu et al., 2020*; *Dou et al., 2020*) were evaluated to validate our analysis (*Figure 7B*).

Progression of the malignant tissue is driven by important alterations in gene expression. Shifts in transcript abundance towards Stage IV were evaluated by differential expression analysis in DESeq2 package (–1> log2FoldChange > 1 and p-value < 0.05), producing a set of 2842 altered genes (*Figure 7C*). Among these differentially expressed genes (DEGs), we found that 102 genes (*Figure 7—source data 1*) were also located in TADs with PgCRs including both upregulated and downregulated genes (*Figure 7D* and *Figure 7—figure supplement 1C and D*). These 102 genes constitute 20% of the 522 protein coding genes in TADs with PgCRs (PgCR-genes; *Figure 7—source data 2*). To test if any randomly picked set of genes could produce the same number of intersected genes as the DEGs, we ran 10,000 iterations of an adapted bootstrapped model in which each randomly generated (without replacement) new set of 2842 genes was contrasted with PgCR-genes. Each intersection was then used to build a distribution (μ71.58) and a 95% confidence interval (2.5%=57; 97.5%=87) that demonstrated that the 102 intersected genes were in fact not part of the normal distribution (*Figure 7E*).

Next, we employed GSEA algorithm to test whether the expression profile of these 102 genes in advanced tumor samples was different to the one observed in Ishikawa cells treated with R5020 or E2. Genes modulated by hormones in Ishikawa cells were mainly regulated in the opposite direction during tumor progression, though most of the genes were not regulated in the conditions of our experiments (*Figure 7F*). Although GSEA results were not statistically robust, genes downregulated in tumor progression were mostly clustered together among genes upregulated by E2 (NES = 1.25; FDR = 0.1669) or R5020 (NES = 0.99; FDR = 0.9547) in Ishikawa cells. On the other hand, genes upregulated in tumor progression were more evenly distributed across treated and untreated Ishikawa samples.

These results indicate that hormones via their receptors maintain a fine balance over the expression levels of these genes, which become disrupted upon tumor progression and that PgCRs regulation involves genes that participate directly or indirectly in key steps of tumor onset and progression.

## Discussion

There seems to be consensus that the way in which combinations of TFs assemble their binding sites contributes to the folding of the genome in cell type specific patterns that orchestrate the physiological coordination of gene expression programs required for the proper development and function of complex organisms (*Lambert et al., 2018*; *Stadhouders et al., 2019*). There is evidence that the same TF can regulate different gene sets in different cell types (*Gertz et al., 2012*), but the mechanisms through which hormone receptors regulate endometrial-specific gene networks had not been previously deciphered. Here, we describe ER alpha and PR binding to the genome of endometrial cancer cells and analyze their specific chromatin context. In this genomic study, we used Ishikawa cells, given that they are a good model of Type I epithelial endometrial cancer (*Nishida, 2002*) containing ERalpha and PR.

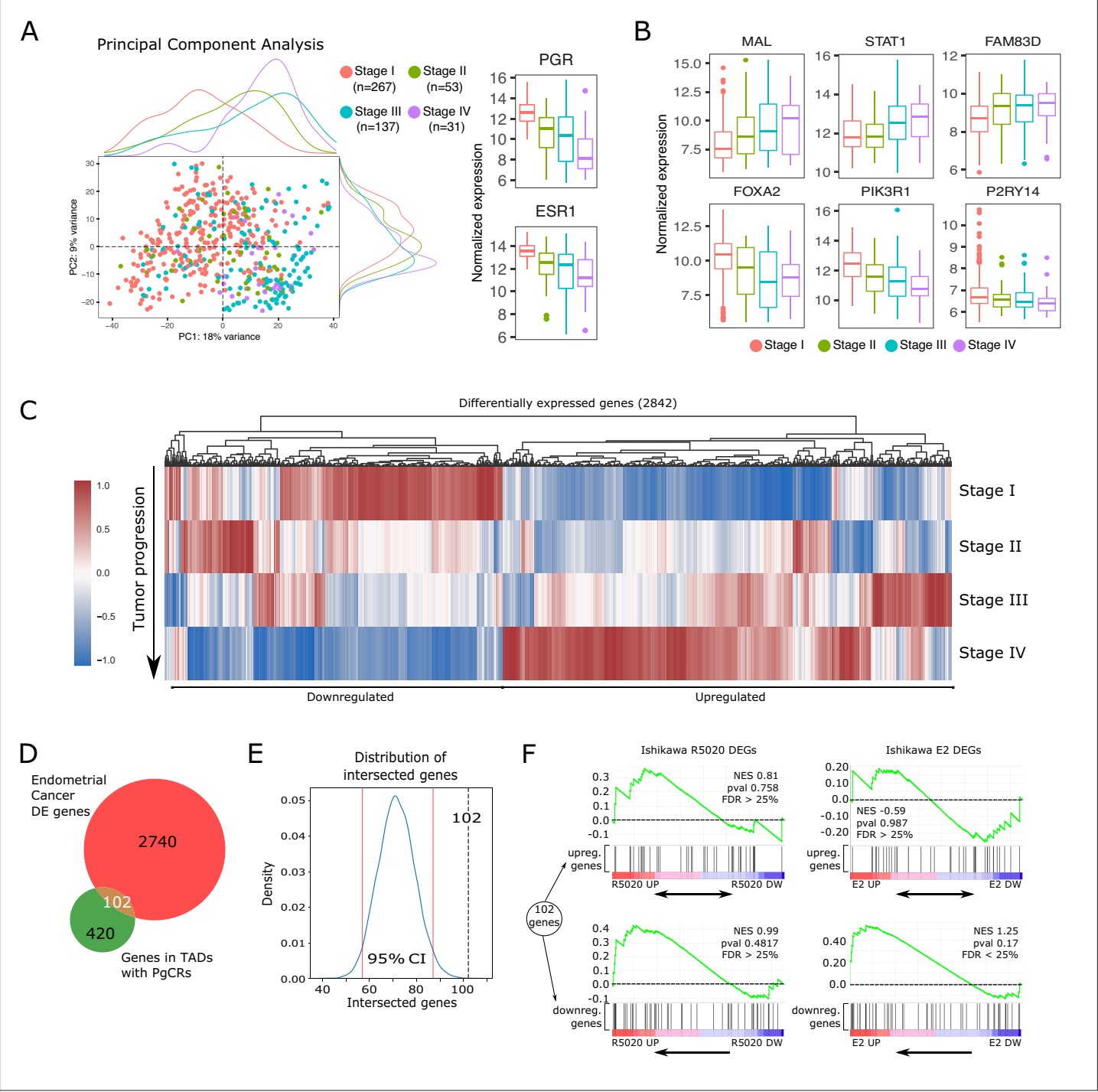

**Figure 7.** Altered expression of PgCR-genes correlates with drivers of endometrial tumor progression. (**A**) Scatter plot of PCA results showing scores of components 1 and 2 (PC1 and PC2) using transcriptomic data of endometrial cancer samples (protein coding genes) obtained from TCGA-UCEC database (n=488). Samples are classified by FIGO stage (I to IV) and identified by color. Marginal density plots represent distribution of scores for each stage. Normalized expression values of PGR and ESR1 genes in these samples is shown to the right of the scatter plot. (**B**) Expression of genes positively (top row) and negatively (bottom row) regulated during endometrial cancer progression (FIGO stage) from TCGA-UCEC samples. (**C**) Heatmap shows scaled normalized counts of 2842 differentially expressed genes (DEGs) between Stage IV and the other three stages. (**D**) Intesection between DEGs presented in C and protein coding genes contained in TADs with PgCRs (522) resulted in 102 gene identities detailed in the box to the left of the Venn diagram. Genes were arranged as Upregulated or Downregulated with respect to Stage IV samples. (**E**) Bootstrapping statistical approach to test whether intersection described in D could result from randomly picking any set of 2842 genes among all known protein coding genes (GENCODE v38). Vertical red lines mark the 2.5 and 97.5 percentile, which denote the 95% confidence interval of the distribution. Intersection from D is indicated as a

*Figure 7 continued on next page*

*Figure 7 continued*

vertical black dashed line (102). (**F**) GSEA plot results using the 102 genes from intersection in D as gene set to match against an expression dataset from Ishikawa cells treated or not with R5020 or E2 for 12h. Normalized enrichment scores and FDR values are reported on the plots.

The online version of this article includes the following source data and figure supplement(s) for figure 7:

**Source data 1.** Intersection between tumor DEGs and PgCR-genes.

**Source data 2.** Protein coding genes in TADs with PgCRs (PgCR-genes).

**Figure supplement 1.** Exploratory analysis and processing of TCGA-UCEC endometrial cancer samples.

It should be noted that progestin regulation of gene expression in Ishikawa cells is decidedly variable. Then, the effects of perturbations to the model (i.e. treatments, siRNA) are tested in a highly variable gene expression background. We would like to remark that the contribution of our results to the progesterone receptor binding and its interaction with their neighboring genomic regions in endometrial cancer cells is novel and more consistent than their contribution to productive gene expression. Therefore, it was relevant to our model to validate the results by contrasting them with previously publish transcriptomic and PR binding data of normal endometrium and endometrial cancer.

It was reported that in Pgr Knockout (PRKO) mice the absence of PR results in unopposed estrogen-induced endometrial hyperplasia (*Lydon et al., 1995*). As for the two isoforms of PR, the PRB isoform is considered a strong transcriptional activator while PRA can function as a transcriptional inhibitor of PRB activity (*Mulac-Jericevic et al., 2000*). Selective ablation of PRA in mice results in a PRB-dependent gain of function, with enhanced estradiol-induced endometrial proliferation (*Conneely et al., 2003*). Ishikawa cells express more PRB than PRA, coherent with PRB dominance in glandular epithelial cells (*Mote et al., 1999*). To explore the mechanism underlying the endometrial specific response to ovarian steroids hormones, we studied the genomic binding of ERalpha and PR by ChIPseq in hormone untreated Ishikawa cells and in cells exposed to hormone for different time periods. We discovered that the majority (67%) of PRbs after estradiol pretreatment were new sites not present in untreated cells and different as well from ERbs occupied after estradiol treatment. Just 639 PR binding sites (11% of all PRbs) were the same for both PR and ERalpha. This indicates that contrary to what was described in breast cancer cells (*Mohammed et al., 2015*; *Singhal et al., 2016*), in endometrial cells PR binding has little influence on ERalpha binding. In Ishikawa cells, binding of ER and PR occurs mainly at ERE and PRE sequences, respectively, in regions that are also enriched in PAX response elements. Ishikawa cells are rich in PAX TF and PAX ChIPseq shows a similar overlapping with ERbs and PRbs.

When we analyzed chromatin topology of Ishikawa cells using Hi-C we found that PRbs and ERbs are enriched in Topologically Associating Domains (TADs) containing hormone-regulated genes. These TADs were predominantly part of the open (A) chromosome compartment, even in cells not exposed to hormone. This was confirmed by ATACseq results showing that the sites where the hormone receptors will bind were already accessible, suggesting that hormone-independent mechanisms were responsible for the generation and maintenance of the hormone-responsive TADs. In that respect, it is interesting that we found an enrichment of PAXbs near PRbs in these TADs containing progesterone regulated genes, suggesting that PAX2 could generate the open chromatin conformation that enables PR binding and facilitates the interacting loops detected in Hi-C experiments. Loss of PAX2 expression has been implicated in the development of endometrial intraepithelial neoplasia (EIN) (*Sanderson et al., 2017*) and PAX2 is potentially useful in the diagnostic of difficult EIN cases (e.g. where there is no 'normal' tissue available to act as an internal control when assessing nuclear morphology) (*Quick et al., 2012*). The fact that PAX2 expression is reduced upon tamoxifen treatment or through neoplastic processes (*Monte et al., 2010*) in which altered hormone receptor pathways are hallmark features is indicative of a central role of the estrogen receptor in its mechanism of action. In line with this notion, we propose that E2-induced ERalpha acts together with PAX2 to define key PR regulatory regions that modulate gene expression. Our results connect PR response elements with PAX2 and 3D chromatin conformation, which is consistent with the preservation of progestin regulation in differentiated cancer cells expressing hormone receptors. Undifferentiated tumor cells, which do not express hormone receptors, lose this regulatory capabilities leading to more aggressive manifestations of the malignancy. We hypothesize that PR-PAX-PR binding sites containing regulatory domains that we name PgCRs could reflect PR shadow enhancers (*Cannavò et al., 2016*) in endometrial cells.

The redundancy of PRbs associated to endometrial specific gene expression may reinforce a genetic mechanism to ensure progestin regulation in tissue under hormonal influence, in periods in which there is low or no circulating hormone. Notably, the only described super-enhancer in endometrial carcinomas is the Myc super-enhancer and is not hormonally regulated (*Zhang et al., 2016*). We postulate the existence of a novel subset of 121 strategic endometrial regulatory domains in this hormonally responsive endometrial cancer cell line. Among them the *TGFA* gene presents one of PgCR-promoter interaction that could explain hormone regulation previously reported in this cells (*Hata et al., 1993*). This concept could be exploited to guide treatments oriented to recover progestin regulation over estrogen proliferative effects in endometrial malignancy.

Previous results in T47D mammary cancer cells have shown Hormone Control Regions, which include ERbs and PRbs acting in conjunction with FOXA1 and C/EBPa (*Nacht et al., 2019*) interact with promoters of hormone regulated genes in hormone-responsive TADs and organize the high level folding of the genome (*Le Dily et al., 2019*). Although the analysis of interaction between PgCR and different ERalpha enriched binding regions in endometrial cells remains to be performed, our present study proposes that PR binding sites originated under estrogenic conditions and acting in conjunction with PAX2, fulfil a similar function in differentiated hormone-responsive endometrial cancer cells. Thus, combinations of the same hormone receptors and different transcription factors account for cell-type-specific expression of different gene regulatory networks in part by generating and maintaining different genome topologies.

*Droog et al., 2017* highlights that 'the divergence between endometrial tumors that arise in different hormonal conditions and shows that ER alpha enhancer use in human cancer differs in the presence of nonphysiological endocrine stimuli'. They reported that ERalpha-binding sites in tamoxifen-associated endometrial tumors are different from those in the tumors from nonusers. It has yet to be explored whether the response to progesterone and sinthetic progestins, used in treatments of hormone-dependent endometrial cancers, is affected by the changes resulting from the use of tamoxifen.

On the other hand, ERalpha and glucocorticoid receptor (GR) are expressed in the uterus and have differential effects on growth (*Vahrenkamp et al., 2018*). Expression of both receptors was associated with poor outcome in endometrial cancer and the simultaneous induction of ER alpha and GR leads to molecular interplay between the receptors (*Vahrenkamp et al., 2018*). In our conditions, R5020 induces genes with GR/PR putative binding sites, enabling regulation that could result in a similar ERalpha-GR pathological outcome.

In the last 10 years an enormous effort has been placed in gathering massive amounts of good quality omic and clinical data from healthy women and endometrial cancer patients in different stages of this often sub-diagnosed disease (*Cancer Genome Atlas Research Network, 2017*; *Dou et al., 2020*). Besides the underlying mutational triggers, the mechanisms by which normal endometrial tissue progresses into aggressive and potentially lethal malignancies are being extensively studied (*Ma et al., 2018*; *Bai et al., 2019*; *Monsivais et al., 2019*; *Rodriguez et al., 2020*). Employing this data, we verified that PAX2, PR and ERalpha levels are reduced in tumor samples (versus normal tissue) and they decreased even further throughout all stages of the disease. In conclusion, it is our belief that loss of PAX2 forces aberrant PR and ER alpha signaling in endometrial cells that may lead to the altered expression of both receptors and consequently PgCR-genes, facilitating the progression of the disease.

## Materials and methods

### Key resources table

| Reagent type (species) or resource | Designation | Source or reference | Identifiers | Additional information |
|---|---|---|---|---|
| Cell line (*Homo sapiens*) | Edometrial adenocarcinoma (Epithelial) Ishikawa | Dr. Rochefort UnitÃ Hormones and Cancer INSERM, France | | Female |
| Cell line (*Homo sapiens*) | FPR Ishikawa | This paper | | Flag-tagged PR overexpression; Female |
| Cell line (*Homo sapiens*) | Breast cancer (Epithelial) T47D | Dr. Beato (Center for Genomic Regulation) | | Female |

*Continued on next page*

*Continued*

| Reagent type (species) or resource | Designation | Source or reference | Identifiers | Additional information |
|---|---|---|---|---|
| Recombinant DNA reagent | p3xFLAG-CMV-14 (plasmid) | Dr. Beato (Center for Genomic Regulation) | | Flag-tagged human PR |
| Sequence-based reagent | Human PAX2 siRNA | Dharmacon | | Pool of 4 on-target oligos |
| Sequence-based reagent | Human control siRNA | QIAGEN 1027310 | | Scramble non-target oligo |
| Antibody | Anti-PR (H-190 Rabbit polyclonal) | Santa Cruz Bio. sc-7208 | RRID:AB_2164331 | ChIP:30μ l xIP Western (1:200) |
| Antibody | Anti-ERalpha (HC-20 Rabbit polyclonal) | Santa Cruz Bio. sc-543 | RRID:AB_631471 | Western (1:200) |
| Antibody | Anti-ERalpha (HC-20X Rabbit polyclonal) | Santa Cruz Bio. sc-543X | | ChIP (25μ l xIP) |
| Antibody | Anti-ERalpha (H-184X Rabbit polyclonal) | Santa Cruz Bio. sc-7204 | | ChIP (25μ l xIP) |
| Antibody | Anti-phosphoserine 294 PR (Rabbit polyclonal) | Cell Signaling 13736 | RRID:AB_2798307 | IF (1:100) |
| Antibody | Anti-PAX2 (Rabbit polyclonal) | BioLegend PRB-276P (Covance) | RRID:AB_291611 | ChIP:6μ l xIP Western (1:200) |
| Antibody | Normal rabbit IgG | Santa Cruz Bio. sc-2027 | RRID:AB_737197 | ChIP (12μ l xIP) |
| Antibody | Anti-alphatubulin (Mouse monoclonal) | Merck T5168 (Sigma-Aldrich) | RRID:AB_477579 | Western (1:1000) |

## Cell culture and hormonal treatments

Endometrial adenocarcinoma Ishikawa cells and FPR Ishikawa cells were cultured in phenol red DMEM/F12 medium (GIBCO, Thermo Fisher Scientific) supplemented with 10% FCS (GreinerBioOne) and gentamycin (Thermo Fisher Scientific) at 37°C and 5% carbon dioxide to maintain cell line stock. Ishikawa cells were obtained from Dr. Rochefort at the Unitá Hormones and Cancer, INSERM U148 et VAC 59 CNRS, 34,090 Montpellier, France. The cell line was authenticated by karyotyping (G band) a sample in comparison with the original published karyotype (*Nishida, 2002*) and has been tested negative for mycoplasma contamination. Before each experiment, cells were plated in phenol red-free DMEM/F12 medium supplemented with 5% dextran-coated charcoal-treated (DCC)-FCS and genta-mycin for 48 hr. Then, the medium was replaced by serum-free DMEM/F12 and kept in it for 18 hr (overnight). Treatments were performed with R5020 and E2 to a final concentration of $10\,nM$ and ethanol (vehicle) for the times indicated for each experiment. When indicated, pretreatment with E2 consisted of a single administration of E2 to a final concentration of $10\,nM$ 12 h before hormonal treat-ments. T47D cells were cultured in RPMI 1640 medium as previously described (*Nacht et al., 2016*).

## Transfection with flag-tagged PR (FPR Ishikawa cells)

Plasmid p3xFLAG-CMV-14 carrying the complete sequence for progesterone receptor gene (HindIII924 - 938EcoRI) was introduced in Ishikawa cells using Lipofectamine 2000 (Thermo Fisher Scientific) following manufacturer recommendations. After 24 hr of transfection, cells were exposed to 0.6 mg/mL G418 for selection. Then on, every two passages, FRP cells were exposed to a reduced concentration of G418 (0.4 mg/mL), except during hormonal treatments.

## Transfection with small interference RNA (SiRNA)

Ishikawa cells were seeded in six well plates ($3 \times 10^5$ per well, western blot and qRTPCR) and p100 plates ($1.5 \times 10^6$ per plate, ChIP) for transfection with pool of 4 on-target siRNAs (Dharmacon; #1: GAAGU-CAAGUCGAGUCUAU, #2: CGACAGAACCCGACUAUGU, #3: GGACAAGAUUGCUGAAUAC, #4: CAUCAGAGCACAUCAAAUC) or scramble non-target (QIAGEN; #1: AATTCTCCGACGTGTCACGT). The following day, cells were transfected using Lipofectamine 3,000 (Thermo Fisher Scientific) following instructions from transfection reagent manufacturer. Briefly, mix A containing OptiMEM and siRNAs and mix B containing OptiMEM and Lipo3000 were incubated together for 20 min at room temperature and added to the cultured in a drop by drop fashion. During incubation, media was replaced by phenol red-free DMEM/F12 5% DCC-FCS without antibiotics. Thirty-six hr later, cells were treated with E2 $10\,nM$ for 12 hr (pretreatment) and then 60 min with R5020 $10\,nM$. Cells were seeded

in three replicates and statistical differences among conditions were evaluated by ANOVA followed by Tukey's HSD and pairwise t test with correction for multiple comparison to produce adjusted p-values (Benjamini-Hochberg method).

## Proliferation assay

Ishikawa cells were seeded at $5 \times 10^4$ cells/plate density in 35-mm dish plates. After 48 hr in 5% DCC-FCS, the medium was replaced for 1% DCC-FCS for 18 hr. Treatments were performed for 48 hr and cells were then collected using trypsin (0.25%). Antagonists for ER and PR, ICI182780 and RU486 1 µM respectively, were added for 60 min and removed before hormonal treatments. The number of live cells was determined using trypan blue (0.1%) in Neubauer chamber, repeating the procedure 16 times for each sample and performing three independent experiments.

## BrdU incorporation assay and cell cycle analysis

Ishikawa cells were seeded and prepared for hormonal treatments as described for Proliferation assay. Treatments were carried out for 15 hr, the last two hours of which includes incubation with BrdU. Cells were treated with cell cycle inhibitor TSA A 250 µM as negative control of BrdU incorporation. After collecting cells in trypsin and washing them with PBS, ethanol 70% was added to fix and permeabilize them. DNA denaturation was achieved with 0.5% BSA and $2\,\mathrm{M}$ HCl after which cells were incubated in 1:2000 solution of anti-BrdU (BD Pharmingen) for 1 hr at RT. FITC secondary antibody (Dako) was incubated for 1 hr in obscurity at RT followed by propidium iodide for 5 min. BrdU incorporation and cell cycle phases were evaluated by flow cytometry (BD FACS Canto II) in three replicates.

## Western blot

Cell extracts were collected at the times indicated by the experiment with 1% SDS, $25\,\mathrm{mM}$ Tris-HCl pH 7.8, $1\,\mathrm{mM}$ EDTA, $1\,\mathrm{mM}$ EGTA and protease and phosphatase inhibitors. Total protein extracts were loaded in 8% SDS-PAGE and incubated with the following antibodies: PR (H190, Santa Cruz Bio.), ERalpha (HC-20, Santa Cruz Bio.) and alpha-tubulin (Sigma Aldrich). Quantification of gel images was performed with ImageJ software and expressed as abundance in relative units to alpha-tubulin.

## Immunofluorescence

Cells were seeded onto coverslips in six-well plates in a density of $10 \times 10^3$ cells/150 µl using the protocol described in Cell culture and hormonal treatments and either pretreated or not with E2 $10\,\mathrm{nM}$ during the last 12 hr of serum-free culture. After hormonal treatments cells were washed with ice cold PBS followed by fixation and permeabilization by incubation in 70% ethanol for 12 hr at -20°C. After rinsing three times for 5 min in 0.1% Tween-PBS, the coverslips were incubated for 2 hr with 10% BSA in 0.1% Tween-PBS to reduce nonspecific staining. To detect PR (H-190 Santa Cruz Bio.), phosphoserine 294 PR (S294 Cell Signaling), ERalpha (HC-20 Santa Cruz Bio.), and PAX2 (Biolegend) cells were incubated with corresponding antibodies diluted in 10% BSA 0.1%Tween-PBS at 4°C overnight. After several washes in Tween-PBS, coverslips were exposed to secondary antibodies Alexa 488 and Alexa 555 (Thermo Fisher Scientific, Thermo Fisher Scientific) diluted 1:1000 in 10% BSA 0.1% Tween-PBS for 1 hr at room temperature using DAPI to reveal nuclei. Coverslips were mounted on slides with Mowiol mounting medium (Sigma Aldrich) and analyzed in TIRF Olympus DSU IX83 (Olympus Life Sciences Solutions). Quantification of nuclear fluorescence was done with ImageJ software after generating a binary mask in dapi images.

## qRTPCR

After 12 h of treatment with R5020 and E2, cell extracts were collected in denaturing solution ($4\,\mathrm{M}$ Guanidine thiocyanate, $25\,\mathrm{mM}$ Sodium citrate pH 7, $0.1\,\mathrm{M}$ 2-Mercaptoethanol, 0.5% Sarkosyl) and total RNA was prepared following phenol-chloroform protocol (*Chomczynski and Sacchi, 1987*). Integrity-checked RNA was used to synthesize cDNA with oligodT (Biodynamics) and MMLV reverse transcriptase (Thermo Fisher Scientific). Quantification of candidate gene products was assessed by real-time PCR. Expression values were corrected by GAPDH and expressed as mRNA levels over time zero (T0). Primer sequences are: *PGR* (FW:5'-CCCACAGGAGTTTGTCAAGC-3' RV:5'-TAACTTCAGACATCATTTC-3'), *ESR1* (FW:5'-TCTATTCCGAGTATGATCCTACCA-3' RV:5'-CAGACGAGACCAATCATCAG-3'), *ESR2* (FW:5'-GTCCTGCTGTGATGAACTAC-3' RV:5'-CCCTCTTTGCGTTTGGACTA-3'),

*ALPP* (FW:5'-CAACCTGAGCTGCCTTTCTC-3' RV:5'-GAACTGTGTCCCGGCTTCT-3'), *TGFA* (FW:5'-CTTCAAGCCAGGTTTTCGAG-3' RV:5'-GGCAGGTTGGAAGAGATCAA-3'), *TIPARP* (FW:5'-CACACCAGCTCACTTCCA GA-3' RV:5'-CAGCTCAAACACGAGGTCAA-3'), *GAPDH* (FW:5'-GAGTCAACGGATTTGGTCGT-3' RV:5'-TTGATTTTGGAGGGATCTCG-3'), *PAX2* (FW:5'-CCAATGGT-GAGAAGAGGAAA-3' RV:5'-CTCAAAGACCCGATCCAAAG-3').

## Luciferase reporter assay

Ishikawa cells were seeded and prepared for hormonal treatments as described for Proliferation assay without addition of gentamycin. Cells were co-transfected with MMTV LTR-Firefly Luciferase (pAGM-MTVLu, gift from Laboratory of Patricia Elizalde) and CMV-Renilla luciferase (pRL-CMV, Promega) plasmids using lipofectamine plus 2000 (Thermo Fisher Scientific). After 5 hr, media were renewed with the addition of antibiotics and 12 hr later cells were treated with vehicle (ethanol) and R5020 for 20 hr. Firefly and Renilla activities (arbitrary units) were determined with Dual-Luciferase Reporter assay system (Promega) and expressed as Firefly units relative to internal control Renilla for each sample (Firefly x10$^4$/Renilla).

## RNAseq

Total RNA was collected from untreated (T0) and 12 hr R5020- and E2-treated Ishikawa cells using RNeasy Plus Mini Kit (QIAGEN) and subjected to high-throughput sequencing in Illumina HiSeq 2000 and 2,500. Poly-A-enriched RNA was used to prepare libraries with TruSeq RNA Sample Preparation kit v2 y v4 (ref. RS-122-2001/2, Illumina) according to instructions from manufacturer followed by single-end (run1) and paired-end (run2) sequencing. Good quality 50 bp reads were aligned to the reference human genome (hg19, UCSC) using Tophat software (*Trapnell et al., 2009*) keeping those that mapped uniquely to the reference with up to two mismatches and transcript assembly, abundance quantification and differential expression analyses were performed with the Cufflinks tool (*Trapnell et al., 2010*). Genes under 200 bp in length or with FPKM values below 0.1 were excluded from downstream analyses. Genes were classified into induced, repressed or non-regulated depending on log2FC value relative to untreated cells (T0). Threshold value was arbitrarily set at log2FC = ±0.8 and q < 0.05 (FDR). Enriched terms and TFBS were determined through RDAVIDWebservice (*Fresno and Fernández, 2013*) and DAVID web-based tool (*Huang et al., 2009*) under standard parameter settings for each tool.

## Gene set enrichment analysis (GSEA)

GSEA tool was implemented following instructions from developers under default parameters (*Subramanian et al., 2005*). The expression dataset was created using Ishikawa RNAseq results, labeling samples as 'R5020' and 'E2' for categorical classification (phenotypes). Gene sets were constructed from proliferative and mid-secretory normal endometrial RNAseq samples (*Chi et al., 2020*). Differential expression analysis to extract genes representative of each stage was performed with DESeq2 package (|log2FC|>2, p < 0.05) (*Love et al., 2014*).

## Chromatin immunoprecipitation (ChIP)

ChIP experiments were performed as described in *Strutt and Paro, 1999* and (*Vicent et al., 2011*). Antibodies used for immunoprecipitation were PR (H-190, Santa Cruz Bio.), ER alpha (HC-20X and H-184X, Santa Cruz Bio.), PAX2 (PRB-276P, BioLegend), and normal rabbit IgG (sc-2027, Santa Cruz Bio.). Enrichment to DNA was expressed as percentage of input (non-immunoprecipitated chromatin, 1%) relative to untreated Ishikawa cells (T0) using the comparative Ct method. Ct values were acquired with BioRad CFX Manager software. Primers sequences are: EGFR UPS (FW:5'-GCGTGAGACACAAACATTCCAAACTGTA-3' RV:5'-GTTCAAGCAATGGGATCGAGTTGT-3'), ALPP UPS (FW:5'-AACTGTTCCAGCTGCGTTTT-3' RV:5'-AGAACACGGTCACTTCCTTGA-3'), ALPP PROM (FW:5'-TGACAGGGTGTCTTGTTCCA-3' RV:5'-GGGT GCGGTATTGAGTACAGA-3'), TGFA UPS (FW:5'-CACATCCGGAGTTCAGACAA-3' RV: 5'-CACCTG-GGAGCAGGTTACTC-3'), TGFA MID (FW:5'-GGCATTTGGAGGGTGTCTAA-3' RV:5'-GAGCAGAGG-GGTCACTGAAG-3'), TGFA P1 (FW:5'-CTCTCACACCAGACGAAGCACA-3' RV:5'-CAGTGACCCCTG AGTTGGAGACT-3'), TGFA PROM (FW:5'-GGGAAAAAGACGCAGACTAGG-3' RV:5'-GGTAGCCG-CCTTCCTATTTC-3'), TIPARP PROM (FW:5'-GAGGCTGGAGGCGTCTGGGGAGTAGG-3' RV:5'-CTGC GGACAGATGGAGGGTCACTTTG-3'), SERPINA3 (FW:5'-GCATCATCAAACTGAAGCCTGAGAA-3'

RV:5'-CAGTAGAAAAGCCTCTTTGTTACTCCCA-3'), RPS6KA1 (FW:5'-GGTACTGTTGTCTGGT CCCCCCT-3' RV:5'-CTCCAGTGAGAACAGCCCAACCT-3').

## ChIPseq

After minor modifications to the ChIP protocol described in *Vicent et al., 2011*, purified ChIP-DNA was submitted to deep sequencing using Illumina HiSeq-2000. Libraries were prepared by the Genomics unit of the CRG Core Facility (Centre for Genomic Regulation, Barcelona, Spain) with NEBNext ChIPseq Library Prep Reagent Set (ref. E6200S, Illumina) and 50 bp sequencing reads were trimmed to remove Illumina adapters and low-quality ends using Trimmomatic (*Bolger et al., 2014*) version 0.33 in single-end mode. Good quality reads were aligned to the reference human genome (hg19 or hg38, UCSC) with BWA (*Li and Durbin, 2009*) v0.7.12 (BWA_MEM algorithm with default parameters) keeping alignments that mapped uniquely to the genome sequence (Samtools version 1.2, *Li and Durbin, 2009*). Overlapping reads were clustered and significant signal enrichments (peaks) were identified by MACS2 v2.1.0 (*Zhang et al., 2008*) using input as background signal. FDR value during initial peak calling steps was set to 0.05 (q), though downstream analyses included only those with q < 10-5. Replication of binding sites was evaluated among treatments (time of exposure to hormone) and conditions (no pretreated, pretreated and FPR) using scatter plots, venn diagrams and heatmaps. Selected sites were validated by qPCR. When necessary peak files were converted to hg38 coordinates using the batch conversion tool from UCSC. Statistical significance of the association between peak files (bed) was evaluated using bedtools (v.2.28.0 *Quinlan, 2014*) shuffle (parameters for randomization: -chrom -noOverlapping) and fisher (pairwise comparisons) modules.

## Heatmaps, scatterplots, and motif analysis

Overlap of ChIPseq peak regions defined by upstream peak calling procedures (MACS2) were determined using intersectBed program from the bedTools suite (*Quinlan, 2014*). An overlap of at least one bp was considered positive. De novo motif discovery (MEME software) performed on sequences contained in 10 kb windows centered in peak summits. Graphs, correlation tests, non-linear regression and statistical analyses in general were performed for common peaks between ChIPseq samples using R (R Development Core Team). Heatmaps were plotted using the summit of the peaks as a reference central position. Reference positions were taken from common and exclusive peaks within experiments and were sorted by height of the peak. Genome aligned reads occurring between -5000 and +5000 bp from reference sites were mapped using count_occurences program (*Kremsky et al., 2015*) and the number of reads per bins of 200 bp was used for the color intensity of heatmap cells with R. For Motif discovery, genomic regions of top 500 peaks ranked by their height were extracted from each set and regions that overlap with repeats, low complexity regions or transposable elements (extracted from the UCSC genome browser, hg19 human assembly), were removed from the analysis. Motif discovery was performed using MEME program suite executed with the following parameters: -maxsize 250000 -revcomp -dna -nmotifs 3 -mod oops (*Bailey et al., 2015*). Motif enrichments were evaluated with the procedure and statistics described in *Agirre et al., 2015*. Additionally, the analysis utilized a 5mers collection of 1395 human position frequency matrices modeling transcription factors binding sites (*Weirauch et al., 2014*), which were scanned (p-value $<1 \times 10^{-4}$) and their enrichment evaluated in regions of 200 bp centered in the summits of whole peaks sets. To uncover motif profiles, discovered and library motifs were whole-genome scanned (p-value$<1 \times 10^{-4}$). Their occurrences around the sets of summits were obtained with count_occurrences (±2000 bp, bin size = 200 bp) and the profiles showing the proportion of regions per bin having at least one match were plotted using R.

## Binding site-gene association

Genomic coordinates of PR and ERalpha binding sites (hg38) were fed to GREAT web tool (*McLean et al., 2010*) to identify potential cis-regulatory interactions. Association was determined in a 'basal plus extension' process using a proximal regulatory domain of 5 kb upstream and 1 kb downstream from each TSS (GRCh38, UCSC hg38) and an extension of 100 kb in both directions. The group of genes associated with PRbs or ERbs were respectively intersected to R5020 and E2 RNAseq results, employing simple python scripting.

## ATACseq

ATACseq was performed as previously described (*Buenrostro et al., 2013*). Briefly, 50,000 cells were lyzed with 50 µL cold lysis buffer (Tris-Cl pH 7.4 10 mM; NaCl 10 mM; MgCl2 3 mM; NP-40 0.1% v/v) and centrifuged at 500xg for 10 min at 4°C. Nuclei were resuspended in TD Buffer with 1.5 µL Tn5 Transposase (Nextera, Illumina) and incubated 15 min at 37°C. DNA was isolated using Qiagen MinElute column and submitted to 10 cycles of PCR amplification using NEBNext High-Fidelity 2 X PCR Master Mix (Univ. primer: AATGATACGGCGACCACCGAGATCTACACTCGTCGGCAGCGTCAGATGTG; Indexed primers: CAAGCAGAAGACGGCATACGAGATNNNNNNNNGTCTCGTGGGCTCGGAGATGT). Library were size selected using AMPure XP beads and sequenced on a NextSeq 500 instrument (2 × 75 nt).

## Hi-C

High-throughput chromosome conformation capture assays were performed as previously described (*Lieberman-Aiden et al., 2009*; *Rao et al., 2014*). Adherent cells were directly cross-linked on the plates with 1% formaldehyde for 10 min at room temperature. After addition of glycine (125 mM final) to stop the reaction, cells were washed with PBS and recovered by scrapping. Cross-linked cells were incubated 30 min on ice in 3 C lysis Buffer (10 mM Tris-HCl pH = 8, 10 mM NaCl, 0.2% NP40, 1 X anti-protease cocktail), centrifuged 5 min at 3000 rpm and resuspended in 190 µL of NEBuffer2 1 X (New England Biolabs - NEB). 10 µL of 10% SDS were added and cells were incubated for 10 min at 65°C. After addition of Triton X-100 and 15 min incubation at 37°C, nuclei were centrifuged 5 min at 3000 rpm and resuspended in 300 µL of NEBuffer2 1 X. Digestion was performed overnight using 400 U MboI restriction enzyme (NEB). To fill-in the generated ends with biotinylated-dATP, nuclei were pelleted and resuspended in fresh repair buffer 1 x (1.5 µL of 10 mM dCTP; 1.5 µL of 10 mM dGTP; 1.5 µL of 10 mM dTTP; 37.5 µL of 0.4 mM Biotin-dATP; 50 U of DNA Polymerase I Large (Klenow) fragment in 300 µL NEBuffer2 1 X). After 45 min incubation at 37°C, nuclei were centrifuged 5 min at 3,000 rpm and ligation was performed 4 hr at 16°C using 10,000 cohesive end units of T4 DNA ligase (NEB) in 1.2 mL of ligation buffer (120 µL of 10X T4 DNA Ligase Buffer; 100 µL of 10% Triton X-100; 12 µL of 10 mg/ml BSA; 963 µL of H2O). After reversion of the cross-link, DNA was purified by phenol extraction and EtOH precipitation. Purified DNA was sonicated to obtain fragments of an average size of 300–400 bp using a Bioruptor Pico (Diagenode; eight cycles; 20 s on and 60 s off). Three µ g of sonicated DNA was used for library preparation. Briefly, biotinylated DNA was pulled down using 20 µL of Dynabeads Myone T1 streptavidine beads in Binding Buffer (5 mM Tris-HCl pH7.5; 0.5 mM EDTA; 1 M NaCl). End-repair and A-tailing were performed on beads using NEBnext library preparation end-repair and A-tailing modules (NEB). Illumina adaptors were ligated and libraries were amplified by 8 cycles of PCR. Resulting Hi-C libraries were first controlled for quality by low sequencing depth on a NextSeq500 prior to higher sequencing depth on HiSeq2000. Hi-C data were processed using an in-house pipeline based on TADbit (*Serra et al., 2017*). Reads were mapped according to a fragment-based strategy: each side of the sequenced read was mapped in full length to the reference genome Human Dec. 2013 (GRCh38/hg38). In the case reads were not mapped when intra-read ligation sites were found, they were split. Individual split read fragments were then mapped independently. We used the TADbit filtering module to remove non-informative contacts and to create contact matrices as previously described (*Serra et al., 2017*) PCR duplicates were removed and the Hi-C filters applied corresponded to potential non-digested fragments (extra-dandling ends), non-ligated fragments (dandling-ends), self-circles, and random breaks.

## CNV

The copy number variation (CNV) analysis was estimated comparing the coverage obtained in the Hi-C datasets with the expected coverage for a diploid genome based on the density of restriction sites and genomic biases (*Vidal et al., 2018*). Indeed, the linear correlation between number of Hi-C contacts and number of restriction sites is lost in case of altered copy number allowing the estimation of a relative number of copy as compared to diploid chromosomes in each dataset. Such estimations are consistent with other analyses and with karyotyping (*Le Dily et al., 2014*).

## Virtual 4C

Hi-C matrices were normalized for sequencing depth and genomic biases using OneD (*Vidal et al., 2018*) and further smoothed using a focal average. Virtual 4 C plots were generated from the matrices locally normalized and expressed as normalized counts per thousands within the region.

### Intra-TAD interactions between specific loci

Each bin of a TAD was labeled as part of a PgCR or TSS (or 'others' if they did not belong to the previous types). We collected the observed contacts between the different types of bins and computed the expected contacts frequencies based on the genomic distance that separate each pair. In the figure, results are expressed as Log2 of the ratio of observed contacts between the different types of pairs above the intra-TAD background.

### Endometrial cancer samples (TCGA)

Raw count data from human endometrial cancer RNAseq samples (n = 575) were downloaded from The Cancer Genome Atlas (TCGA), project TCGA-UCEC. Each sample was matched to its corresponding clinical metadata including FIGO stage and histologic type and only protein coding genes above arbitrary threshold (mean expression among samples > 100 counts) were kept for further analyses. Raw counts were normalized and transformed (variance stabilization) in DESeq2 package and later used for filtering Stage I samples by *ESR* and *PGR* expression levels. Considering the inherently heterogeneous nature of tissue samples, only samples clustered together above threshold were kept for further analysis (see *Figure 7—figure supplement 1A*). After filtering, the raw counts of 488 samples were used to find differential expressed genes (DEG) between stages, downstream heatmaps (pheatmap R package *Kolde and Kolde, 2015*) and Principal Component Analysis in DESeq2 and PCAtools packages (*Blighe et al., 2019*). To construct the normal distribution of intersected genes we randomly sampled (without replacement) a group of the same size of DEGs among all possible protein coding genes (GENCODE v38) 10,000 times using custom python scripting (code is publicly available at https://github.com/SaraguetaLab/ishikawa_scripting, copy archived at swh:1:rev:25a4757c-c21053544bbc45144f34d3f033e7e7d2, *La Greca, 2021*), then intersected each one with PgCR-genes to produce the curve. Statistical significance of our non-random intersection was determined by calculating the 2.5 and 97.5 percentiles in the distribution (95% confidence interval).

## Acknowledgements

We are grateful to members of the Beato and Saragüeta laboratories for help and suggestions.

## Additional information

### Funding

| Funder | Grant reference number | Author |
|---|---|---|
| Consejo Nacional de Investigaciones Científicas y Técnicas | PIP 2015-682 | Patricia Saragüeta |
| Fondo para la Investigación Científica y Tecnológica | PICT 2015-3426 | Patricia Saragüeta |
| H2020 European Research Council | FP7/2007-2013 grant agreement 609989 | Miguel Beato |

The funders had no role in study design, data collection and interpretation, or the decision to submit the work for publication.

### Author contributions

Alejandro La Greca, Conceptualization, Data curation, Formal analysis, Investigation, Methodology, Visualization, Writing – original draft, Writing – review and editing; Nicolás Bellora, Data curation, Formal analysis, Methodology, Visualization, Writing – original draft; François Le Dily, Data curation, Formal analysis, Methodology, Visualization, Writing – original draft, Writing – review and editing; Rodrigo Jara, Javier Quilez Oliete, Data curation, Formal analysis, Visualization; Ana Silvina Nacht, Formal analysis, Methodology; José Luis Villanueva, Enrique Vidal, Data curation, Formal analysis, Methodology, Visualization; Gabriela Merino, Cristóbal Fresno, Data curation, Formal analysis; Inti

Tarifa Reischle, Methodology; Griselda Vallejo, Elmer Fernández, Formal analysis, Investigation, Supervision; Guillermo Vicent, Miguel Beato, Investigation, Supervision, Writing – review and editing; Patricia Saragüeta, Conceptualization, Formal analysis, Funding acquisition, Investigation, Project administration, Supervision, Writing – original draft, Writing – review and editing

**Author ORCIDs**
Alejandro La Greca [ID] http://orcid.org/0000-0002-0309-7683
Nicolás Bellora [ID] http://orcid.org/0000-0001-6637-3465
Patricia Saragüeta [ID] http://orcid.org/0000-0001-8222-9690

**Decision letter and Author response**
Decision letter https://doi.org/10.7554/eLife.66034.sa1
Author response https://doi.org/10.7554/eLife.66034.sa2

---

## Additional files

### Supplementary files
• Transparent reporting form

### Data availability
All raw and processed sequencing data generated in this study have been submitted to the NCBI Gene Expression Omnibus under accession number GSE139398. Source data file has been provided for Figure 6. T47D ChIPseq data is available under GEO accession number GSE41466 (Ballare et al, 2013) and Hi-C data in GEO accession GSE53463 (Le-Dily et al, 2014). RNAseq datasets from proliferative (GSM3890623, GSM3890624, GSM3890625 and GSM3890626) and mid-secretory (GSM3890627, GSM3890628, GSM3890629, GSM3890630 and GSM3890631) human endometrium were obtained from GEO accession GSE132711 (SuperSeries GSE132713) (Chi et al, 2020). ChIPseq coverage data of proliferative and secretory normal endometrium were downloaded from GEO accession GSE132712 (SuperSeries GSE132713) (Chi et al, 2020). Human endometrial cancer RNAseq samples (n=575) were downloaded from The Cancer Genome Atlas (TCGA), project TCGA-UCEC. Additional normal and endometrial cancer samples (n=109) were accessed through CPTAC program in the National Cancer Institute using cptac platform installed with python (Dou et al, 2020).

The following dataset was generated:

| Author(s) | Year | Dataset title | Dataset URL | Database and Identifier |
|---|---|---|---|---|
| La Greca A, Bellora N, Le Dily F, Jara R, Quilez Oliete J, Villanueva JL, Vidal E, Merino G, Fresno C, Vallejo G, Vicent GP, Fernández E, Beato M, Saragüeta P, Rieschle T | 2019 | Higher-order chromatin organization defines PR and PAX2 binding to regulate endometrial cancer cell gene expression | https://www.ncbi.nlm.nih.gov/geo/query/acc.cgi?acc=GSE139398 | NCBI Gene Expression Omnibus, GSE139398 |

The following previously published datasets were used:

| Author(s) | Year | Dataset title | Dataset URL | Database and Identifier |
|---|---|---|---|---|
| Chi RA, Wang T, Adams N, Young SL, Spencer TE, DeMayo F, Wu SP | 2019 | Endometrial transcriptome and PGR cistrome in cycling fertile women | https://www.ncbi.nlm.nih.gov/geo/query/acc.cgi?acc=GSE132713 | NCBI Gene Expression Omnibus, GSE132713 |

*Continued on next page*

*Continued*

| Author(s) | Year | Dataset title | Dataset URL | Database and Identifier |
|---|---|---|---|---|
| Le Dily F, Baù D, Pohl A, Vicent G, Soronellas D, Castellano G, Serra F, Wright RH, Ballare C, Filion G, Marti-Renom MA, Beato M | 2014 | Distinct structural transitions of chromatin topological domains coordinate hormone-induced gene regulation | https://www.ncbi.nlm.nih.gov/Traces/study/?acc=PRJNA232055&o=acc_s%3Aa | NCBI Gene Expression Omnibus, GSE53463 |
| Nacht SA, Pohl A, Zaurin R, Soronellas D, Quilez J, Sharma P, Wright RH, Beato M, Vicent GP | 2016 | Hormone induced repression of genes requires BRG1-mediated H1.2 deposition at target promoters | https://www.ncbi.nlm.nih.gov/Traces/study/?acc=PRJNA326976&o=acc_s%3Aa | NCBI Gene Expression Omnibus, GSE83785 |
| Ballare C, Castellanos G, Gaveglia L, Althammer S, Gonzalez-Vallinas J, Eyras E, Zaurin R, Soronellas D, Vicent G, Beato M | 2012 | Nucleosome driven transcription factor binding and gene regulation | https://www.ncbi.nlm.nih.gov/geo/query/acc.cgi?acc=GSE41466 | NCBI Gene Expression Omnibus, GSE41466 |

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
