## [Editor Report]

This study modeling the actions of estrogen and progesterone receptors (ER and PR) in endometrial cancer cells through a panel of genomic approaches reveals a potentially interesting collaboration between the two, further facilitated by the non-receptor transcription factor PAX2. The identification of so-called chromatin 'progestin control regions' inside TADs, where the three factors cooperate and which appear to be the feature setting endometrial cancer cells apart from breast cancer cells, is of potential interest for future investigation.

---

## [Decision Letter]

**Decision letter after peer review:**

Thank you for submitting your article "Chromatin topology defines estradiol-primed Progesterone Receptor and PAX2 binding in endometrial cancer gene expression" for consideration by *eLife*. Your article has been reviewed by 3 peer reviewers, and the evaluation has been overseen by a Reviewing Editor and Jessica Tyler as the Senior Editor. The following individual involved in review of your submission has agreed to reveal their identity: Carol Lange (Reviewer #2).

Essential revisions:

1) The study is devoid of any functional data linking PAX2 to ER/PR signaling. Co-localization data is purely correlative and circumstantial. PAX2 binding enrichment near ER/PR sites over its binding at randomly selected open enhancers needs to be statistically established. Critically, the authors should demonstrate that depleting PAX2 affects ER/PR-driven gene expression or their genomic localization. As a related point, the authors should specify how they arrived at picking PAX2 is a candidate regulator among other members of the Pax family.

2) Although Ishikawa cells are a well-studied commonly used model system for endometrial cancer cells, they are a unique cell line and it is critical to establish that authors' findings are not a feature specific to Ishikawa cells. The authors need to recapitulate at least some of their observations in a different endometrial ER+/PR+ model cell lines, or human tissue organoids, or in cohorts of clinical samples. If no other models exist, the authors should acknowledge and discuss the limitations of their findings.

3) Based on the GEO dataset uploaded, there appears to be a single replicate for the ChIP-seq experiments. If so, this does not meet the minimum ENCODE requirement and cannot be analyzed or serve as basis for any conclusions. For example, many of the differential peaks (i.e. the 307 lost peaks) are likely to be false positives that result from only one replicate.

4) Many of the conclusions are based on small subsets of genes in TADs that correlate with outcome. The authors need to provide evidence that the Hi-C and genomic analyses revealed a statistically enriched gene signature of clinical and prognostic potential in endometrial cancer, rather than pick a handful of genes that happen to correlate with clinical outcome (which would likely be achieved by any random large enough starting set of genes).

Additional specific revisions:

1. Based on motif enrichment analysis, the authors claim that PR monomers could bind at 30 min and dimers at 60 min. There is no real evidence that this is the case. Unless the authors plan to pursue this functionally and show dimeric vs. monomeric binding, this statement needs to be removed.

2. Please clarify what 'shuffling the coordinates' for PgCRs means (lines 362-363)

3. On page 13, the term 'hormone-regulated' is used frequently. Please state whether you are referring to R5020 or E2.

3. In line 398, please clarify what list of genes you are referring to.

4. Some of the text is confusing. For example, the authors talk about the 'proliferative response' to R5020, which implies a proliferation in response to treatment, rather than an anti-proliferative response. Similarly, the sentence starting 'The signature of genes regulated in conjunction with loss….' is unnecessarily confusing and doesn't make obvious sense.

*Reviewer #1 (Recommendations for the authors):*

1. Some functional data linking PAX2 to the PR (and/or) ER pathway.

2. Some confidence that more than one replicate of ChIP-seq was conducted and if not, I am unwilling to support pursuing this manuscript.

3. Evidence that the Hi-C and broad genomic analysis revealed a gene signature of clinical value, rather than a handful of genes that happen to correlate with clinical outcome (which would likely be achieved any random set of genes).

I found the final part confusing and not convincing. Ultimately the key message is: 1. PR is a prognostic gene; 2. A small number of the genes from the TAD analysis correlate with outcome. This would be expected if enough genes are used as a starting point. Is there any statistical enrichment in the genes derived from the Hi-C (assuming they are also regulated in the authors RNA-seq data) that correlate with outcome in endometrial cancer? If not, then any list of genes would inevitably have a handful that correlate with outcome, but the key question is whether the Hi-C compartmentalization allows the discovery of genes that are enriched for clinical prognostic potential? If not, I'm not sure the last part adds anything to the story.

*Reviewer #2 (Recommendations for the authors):*

The authors use Ishikawa cells as a representative cell line model of PR+ endometrial cancer. Are there additional ER+/PR+ models than can be included to support their salient findings around the role of PAX2 as a required co-factor with PR or ER/PR at hormone regulated target genes? The major concern for readers centers on how we dissociate cell line differences from tissue-specific ones with the use of only one cell line to represent the tissue? Can the authors demonstrate a role for PAX2 in human tissue organoids for example? If no other models exist, the authors should discuss the limitations of their findings that are limited to a single cell line herein.

How does PAX2 expression change during endometrial cancer development and progression (i.e. in the public data)? Loss of PAX2 (i.e. early in cancer development) may explain the loss of responsiveness to progesterone or loss of protection from progesterone as an inhibitor of estrogen-induced proliferation. Furthermore, the authors should demonstrate that PAX2 is actually required for progestin-dependent regulation of known target genes in these cells by performing PAX2 knock-down or knock-out studies (and by individual target gene readouts of mRNA levels and promoter recruitment/ChIP).

*Reviewer #3 (Recommendations for the authors):*

1) The author claims that ER alpha is the predominant isoform present in Ishikawa cells (Supplementary Figure 1G). This relies on a Western blot comparing abundance of ER alpha to ER beta. However, this Western has no positive control for the ER beta antibody, and given different sensitivities of the antibodies, abundance between the two isoforms cannot really be compared in this manner. Has this been looked at previously? Is ER beta not expressed at all in this cell line?

2) There needs to be more rationale for the use of the Ishikawa cell line. As per the authors' data, there is very low PR expression in these cells, and the luciferase activity of the endogenous PR / number of PR chromatin binding sites is overall low throughout the ChIP-seq studies. The authors also state on line 262 that the hormone-dependent gene regulation is very cell type specific. Why not look at common themes across cell lines instead of looking very specifically at a cell line for which the data likely will not apply to other cell lines?

3) Figure 4 shows many other enriched TF binding motifs other than PAX family TFs. It would be useful to discuss why the authors chose to focus on PAX family/PAX2 specifically.

4) On lines 297-307, the authors state that PAX2 ChIP-seq data shows overlap with ER and PR binding sites. However, do the PAX2 binding sites specific ally correspond to new PR binding sites which appear in the presence of E2 pre-treatment?

5) Can the authors be more clear on what they mean by shuffling the coordinates for PgCRs in lines 362-363?

6) On page 13, the authors use the term "hormone-regulated" frequently. Please clarify this language to express whether you are referring to progestin or E2 regulated genes.

7) In lines 398, it is unclear what list of genes the authors are referring to. The data generated in this study have been submitted to the NCBI Gene Expression Omnibus (GEO); accession number GSE139398.

[Editors' note: further revisions were suggested prior to acceptance, as described below.]

Thank you for resubmitting your work entitled "Chromatin topology defines estradiol-primed Progesterone Receptor and PAX2 binding in endometrial cancer" for further consideration by *eLife*. Your revised article has been evaluated by Jessica Tyler (Senior Editor) and a Reviewing Editor.

The manuscript has been improved but there are some remaining issues that need to be addressed, as outlined below:

1. The statistical analysis for some of the new figure panels is lacking. Specifically, Figure 5H, 5I, Figure 5—figure supplement 2B (siRNA KD of PAX2) display error bars, and up/down changes are described in Results, however, it is not indicated which tests were used to compare the values and which of the changes are, in fact, significant.

2. Wording in the abstract describing these results could be streamlined to "…. in PAX2 knockdown cells suggests a role for PAX2 in fine-tuning ERalpha and PR interplay in transcriptional regulation".

---

## [Author Response]

Essential revisions:1) The study is devoid of any functional data linking PAX2 to ER/PR signaling. Co-localization data is purely correlative and circumstantial. PAX2 binding enrichment near ER/PR sites over its binding at randomly selected open enhancers needs to be statistically established. Critically, the authors should demonstrate that depleting PAX2 affects ER/PR-driven gene expression or their genomic localization. As a related point, the authors should specify how they arrived at picking PAX2 is a candidate regulator among other members of the Pax family.

Functional data linking PAX2 to ER/PR signaling were studied using depletion of PAX2 with specific siRNA, showing a clear effect on hormone-regulated genes. These effects were observed in both E2 pretreated and non-pretreated cells, indicating that PAX2 is involved in many aspects of PR regulatory action and its interplay with ERalpha. Results on PR binding and gene expression with or without siRNA against PAX2 have been included in new Figure 5 (line 329 of revised manuscript). The Materials and methods and discussion were also revised accordingly.

As co-localization data could be circumstantial, we evaluated the coverage signal of PR and ERalpha ChIP-seq data on a randomized set of PAX2 peaks (shuffled regions). To do so, we employed the very well-known bedtools module “bedtools shuffle” which preserves information from input file, such as peak size (range of coordinates) and chromosome. Also, we added the results from the module “bedtools fisher” in which we tested the significance of the overlap (paragraph starts in line 314 of revised manuscript).

We selected PAX2 as a candidate regulator among other members of the PAX family because there is evidence that loss of PAX2 correlates with a bad prognosis of endometrial tumors (EIN) (see Sanderson et al., 2017 in revised manuscript). Also, because there is an enrichment of PAX2 specific DNA response elements in close proximity with PR and ERalpha binding sites and is predicted to bind in genes regulated by both hormones, unlike the other detected PAX family members or TFs (Figure 4). New analyses included in the revised version of our manuscript will also demonstrate that PAX2 is not only significantly downregulated at the onset of endometrial carcinomas but it is involved in PR-ERalpha target gene regulation as well (Figure 5).

2) Although Ishikawa cells are a well-studied commonly used model system for endometrial cancer cells, they are a unique cell line and it is critical to establish that authors' findings are not a feature specific to Ishikawa cells. The authors need to recapitulate at least some of their observations in a different endometrial ER+/PR+ model cell lines, or human tissue organoids, or in cohorts of clinical samples. If no other models exist, the authors should acknowledge and discuss the limitations of their findings.

As the reviewers comment, there are not well established endometrial cell lines with Ishikawa characteristics, and considering that the use of a single cell line is in fact a limitation because there is a possibility that findings are an Ishikawa specific feature, we compared our transcriptomic data to normal endometrial gene expression profiles obtained from publicly available data (Chi et al., 2020). These results were included in Figure 1, in which we studied the resemblance of normal mid-secretory and proliferating endometrium with hormone-treated or not Ishikawa cells. Also, we determined that the PR binding profile of normal endometrium samples (Chi et al., 2020) was mostly reproduced by our PR ChIPseq samples, particularly in the case of E2 pretreated Ishikawa cells (see new Figure 2—figure supplement 2H). By performing these experiments, we showed that treated Ishikawa cells evince the major characteristics of cycling endometrial cells. In addition, we included analyses on publicly available transcriptomic samples from TCGA-UCEC project and CPTAC (Dou et al. 2020) to validate the relevance of our findings. These are shown in Figures 5 and 7.

The limitations of our findings have now been discussed (line 495).

3) Based on the GEO dataset uploaded, there appears to be a single replicate for the ChIP-seq experiments. If so, this does not meet the minimum ENCODE requirement and cannot be analyzed or serve as basis for any conclusions. For example, many of the differential peaks (i.e. the 307 lost peaks) are likely to be false positives that result from only one replicate.

We processed data from replicates and found similar results between them (see Figure 2—figure supplement 2). While PRbs in E2-pretreated cells and ERbs were mostly reproduced (80% and 70%, respectively), PRbs in non-pretreated cells showed 40% of common peaks. This is mainly due to the fact that one of the replicates produced a higher number of peaks above threshold during peak calling, forcing a lower percentage of common peaks. However, we have demonstrated that almost all reported PRbs were independently reproduced by PR ChIP-seq data in other conditions, namely E2-pretreated PRbs and PRbs from FPR cells (Figure 2 and Figure 2—figure supplement 2). It is important to note that only PRbs from E2-pretreated cells and ERbs promoted the main conclusions of our manuscript. We will upload new data to GEO as soon as possible.

4) Many of the conclusions are based on small subsets of genes in TADs that correlate with outcome. The authors need to provide evidence that the Hi-C and genomic analyses revealed a statistically enriched gene signature of clinical and prognostic potential in endometrial cancer, rather than pick a handful of genes that happen to correlate with clinical outcome (which would likely be achieved by any random large enough starting set of genes).

Using Hi-C coupled with ChIP-seq and ATAC-seq data we demonstrated that PgCRs are in close contact with gene promoters, suggesting a role as potential master regulators. However, we redefined the following analyses to avoid biased conclusions. We extracted genes differentially expressed at different stages of progression from a cohort of endometrial cancer samples from TCGA-UCEC. This set included several genes from TADs with PgCRs, many more than what is expected by chance. Also, many of these genes were regulated by R5020 and/or E2 in the opposite direction (see Figure 7 in revised manuscript).

Additional specific revisions:1. Based on motif enrichment analysis, the authors claim that PR monomers could bind at 30 min and dimers at 60 min. There is no real evidence that this is the case. Unless the authors plan to pursue this functionally and show dimeric vs. monomeric binding, this statement needs to be removed.

The statement has been removed.

2. Please clarify what 'shuffling the coordinates' for PgCRs means (lines 362-363)

Shuffling refers to the use of the bedtools (toolkit for genomic coordinates analysis) module “shuffle”. This tool generates a randomized version of the input coordinates preserving information such as peak size (range of coordinates) and chromosome. Therefore, our analysis aimed to verify if ATAC-seq signal was visible on randomly rearranged coordinates with the same characteristics of PgCRs. The statement has been clarified in the revised manuscript (see line 410).

3. On page 13, the term 'hormone-regulated' is used frequently. Please state whether you are referring to R5020 or E2.

The term is purposely broad to include both hormones in the statement given that particular set of analyses was performed on all regulated genes.

3. In line 398, please clarify what list of genes you are referring to.4. Some of the text is confusing. For example, the authors talk about the 'proliferative response' to R5020, which implies a proliferation in response to treatment, rather than an anti-proliferative response. Similarly, the sentence starting 'The signature of genes regulated in conjunction with loss….' is unnecessarily confusing and doesn't make obvious sense.

The list is no longer valid as this analysis was removed from the revised version of the manuscript. Please, see the Results section connected to Figure 7.

Reviewer #1 (Recommendations for the authors):1. Some functional data linking PAX2 to the PR (and/or) ER pathway.

Functional role of PAX2 was evaluated using a specific siRNA. Please refer to point 1 of the Essential revisions above.

2. Some confidence that more than one replicate of ChIP-seq was conducted and if not, I am unwilling to support pursuing this manuscript.

ChIP-seq replicates were performed for PR, ERalpha and E2 pre-treated PR treated 60min with the corresponding hormone. Please refer to point 3 of the Essential revisions above.

3. Evidence that the Hi-C and broad genomic analysis revealed a gene signature of clinical value, rather than a handful of genes that happen to correlate with clinical outcome (which would likely be achieved any random set of genes).I found the final part confusing and not convincing. Ultimately the key message is: 1. PR is a prognostic gene; 2. A small number of the genes from the TAD analysis correlate with outcome. This would be expected if enough genes are used as a starting point. Is there any statistical enrichment in the genes derived from the Hi-C (assuming they are also regulated in the authors RNA-seq data) that correlate with outcome in endometrial cancer? If not, then any list of genes would inevitably have a handful that correlate with outcome, but the key question is whether the Hi-C compartmentalization allows the discovery of genes that are enriched for clinical prognostic potential? If not, I'm not sure the last part adds anything to the story.

We agree that the key question is that the Hi-C compartmentalization allows the discovery of genes that are enriched for clinical prognostic potential. Then, we have revised the analysis in Figure 7 to avoid biased conclusions. These results are shown in new Figure 7. We found 102 genes associated with progression of endometrial cancer. Bootstrapping analysis ensures that this number of genes is higher than random sample distribution (Figure 7E). Not only mRNA encompassed in PR binding TADs could be relevant to deregulation in cancer but also other regulatory elements (like enhancers, insulators, tethers) present in the same domains to be studied in more detail. Then, the last part of the manuscript is critical to understand endometrial progesterone regulation.

Reviewer #2 (Recommendations for the authors):The authors use Ishikawa cells as a representative cell line model of PR+ endometrial cancer. Are there additional ER+/PR+ models than can be included to support their salient findings around the role of PAX2 as a required co-factor with PR or ER/PR at hormone regulated target genes? The major concern for readers centers on how we dissociate cell line differences from tissue-specific ones with the use of only one cell line to represent the tissue? Can the authors demonstrate a role for PAX2 in human tissue organoids for example? If no other models exist, the authors should discuss the limitations of their findings that are limited to a single cell line herein.

Please, refer to point 2 of the Essential revisions above.

How does PAX2 expression change during endometrial cancer development and progression (i.e. in the public data)? Loss of PAX2 (i.e. early in cancer development) may explain the loss of responsiveness to progesterone or loss of protection from progesterone as an inhibitor of estrogen-induced proliferation. Furthermore, the authors should demonstrate that PAX2 is actually required for progestin-dependent regulation of known target genes in these cells by performing PAX2 knock-down or knock-out studies (and by individual target gene readouts of mRNA levels and promoter recruitment/ChIP).

In silico PAX2 expression from publicly available endometrial cancer data were analyzed (TCGA-UCEC project and CPTAC). These results are represented in Figure 5 and 7. Please, refer to point 1 of the Essential revisions above for further details on this matter.

Reviewer #3 (Recommendations for the authors):1) The author claims that ER alpha is the predominant isoform present in Ishikawa cells (Supplementary Figure 1G). This relies on a Western blot comparing abundance of ER alpha to ER beta. However, this Western has no positive control for the ER beta antibody, and given different sensitivities of the antibodies, abundance between the two isoforms cannot really be compared in this manner. Has this been looked at previously? Is ER beta not expressed at all in this cell line?

The Reviewer raises a valid point regarding the absence of a positive control in the experiment. However, it is important to clarify that this figure represents an end-point PCR with specific primers for each gene and not a western blot as it was argued. Moreover, both sets of primers have been previously used in publications from our laboratory (Vallejo et al., 2005) and further experiments involving ERalpha were performed with well-known antibodies that accurately discriminate the proteins. Nevertheless, we consider the Reviewer’s conceptual remark to be accurate, so we will not oppose to remove it altogether as we believe it conveys no major issue to the manuscript’s main conclusions.

2) There needs to be more rationale for the use of the Ishikawa cell line. As per the authors' data, there is very low PR expression in these cells, and the luciferase activity of the endogenous PR / number of PR chromatin binding sites is overall low throughout the ChIP-seq studies. The authors also state on line 262 that the hormone-dependent gene regulation is very cell type specific. Why not look at common themes across cell lines instead of looking very specifically at a cell line for which the data likely will not apply to other cell lines?

Please, refer to point 2 of the Essential revisions above.

3) Figure 4 shows many other enriched TF binding motifs other than PAX family TFs. It would be useful to discuss why the authors chose to focus on PAX family/PAX2 specifically.

Many other TF binding motifs were enriched in progestin and estradiol regulated genes shown in Figure 4. PAX2 was selected as a candidate regulator among other members of the PAX family because PAX2 response element was the only one found overrepresented in both E2 and R5020 regulated genes and because there is evidence that loss of PAX2 correlates with a bad prognosis of endometrial tumors (EIN) (see Sanderson et al., 2017 in revised manuscript). Please, refer to point 1 of the Essential revisions above for further details on this matter.

4) On lines 297-307, the authors state that PAX2 ChIP-seq data shows overlap with ER and PR binding sites. However, do the PAX2 binding sites specific ally correspond to new PR binding sites which appear in the presence of E2 pre-treatment?

Our results show that PAX2 cooperates with PR binding in the absence of pretreatment with E2 (Figure 5), though we found that it preferentially co-localizes with PRbs under estrogenic conditions. Comparison between PAX2 enrichment and PRbs in non-pretreated and pretreated cells is included in Figure 5D, E and F.

5) Can the authors be more clear on what they mean by shuffling the coordinates for PgCRs in lines 362-363?

Please, refer to point 2 of the Additional specific revisions above.

6) On page 13, the authors use the term "hormone-regulated" frequently. Please clarify this language to express whether you are referring to progestin or E2 regulated genes.

Please, refer to point 3 in the Additional specific revisions section above.

7) In lines 398, it is unclear what list of genes the authors are referring to. The data generated in this study have been submitted to the NCBI Gene Expression Omnibus (GEO); accession number GSE139398.

This section of the analysis has been removed from the revised manuscript. Please, refer to the new version of the text connected with Figure 7 for further details.

[Editors' note: further revisions were suggested prior to acceptance, as described below.]

The manuscript has been improved but there are some remaining issues that need to be addressed, as outlined below:1. The statistical analysis for some of the new figure panels is lacking. Specifically, Figure 5H, 5I, Figure 5—figure supplement 2B (siRNA KD of PAX2) display error bars, and up/down changes are described in Results, however, it is not indicated which tests were used to compare the values and which of the changes are, in fact, significant.

Statistical analysis has now been properly included in the figure panels as well as in legend of these figures. We performed ANOVA statistical analysis followed by Tukey’s HSD on delta Ct values in both datasets (included as Figure 5-Source data1 and 2) as they better modeled data structure (non-transformed values tend to show biased results favoring large differences). Adjusted p-values indicate significant statistical relations between conditions (multiple pairwise comparisons). Also, to improve visualization of panels in Figure 5 and Figure 5-Figure Supplement 2, we included all values in the dataset as dots (see Figure 5H and I and Figure 5—figure supplement 2B of revised manuscript).

2. Wording in the abstract describing these results could be streamlined to "…. in PAX2 knockdown cells suggests a role for PAX2 in fine-tuning ERalpha and PR interplay in transcriptional regulation".

The sentence has been corrected in the revised version of our manuscript.